# Structural insights into photosystem II supercomplex and trimeric FCP antennae of a centric diatom *Cyclotella meneghiniana*

Songhao Zhao[1,6], Lili Shen[1,2,6], Xiaoyi Li[1,3,6], Qiushuang Tao[1,2,6], Zhenhua Li[1,2], Caizhe Xu[1,4], Cuicui Zhou[1,2], Yanyan Yang[1,3], Min Sang[3], Guangye Han [1,3], Long-Jiang Yu [1,3], Tingyun Kuang [1,3], Jian-Ren Shen [1,3,5] ✉ & Wenda Wang [1,3] ✉

Diatoms are dominant marine algae and contribute around a quarter of global primary productivity, the success of which is largely attributed to their photosynthetic capacity aided by specific fucoxanthin chlorophyll-binding proteins (FCPs) to enhance the blue-green light absorption under water. We purified a photosystem II (PSII)-FCPII supercomplex and a trimeric FCP from *Cyclotella meneghiniana* (Cm) and solved their structures by cryo-electron microscopy (cryo-EM). The structures reveal detailed organizations of monomeric, dimeric and trimeric FCP antennae, as well as distinct assemblies of Lhcx6_1 and dimeric FCPII-H in PSII core. Each Cm-PSII-FCPII monomer contains an Lhcx6_1, an FCP heterodimer and other three FCP monomers, which form an efficient pigment network for harvesting energy. More diadinoxanthins and diatoxanthins are found in FCPs, which may function to quench excess energy. The trimeric FCP contains more chlorophylls *c* and fucoxanthins. These diversified FCPs and PSII-FCPII provide a structural basis for efficient light energy harvesting, transfer, and dissipation in *C. meneghiniana*.

Oxygenic photosynthetic organisms play fundamental roles in maintaining the global biosphere by producing carbohydrates and molecular oxygen required for sustaining almost all life forms on the earth. After endosymbiosis of cyanobacteria, oxygenic photosynthetic organisms diverge into two main branches, namely, green and red lineages. Diatoms are one of the main players in the red lineage[1]. They are unicellular photosynthetic organisms originating from secondary endosymbiosis and contribute around 25% of the global photosynthetic net primary production[2,3]. All oxygenic photosynthetic organisms including diatoms have conservative photosynthetic light-energy conversion machinery, photosystem I (PSI) and II (PSII). One of the important factors responsible for the powerful vitality of diatoms is

their extraordinary light-harvesting capacity, which is achieved by their unique antenna system, fucoxanthin (Fx) chlorophyll (Chl) *c* binding proteins (FCPs)[4–6]. Compared with the light-harvesting complexes (LHCI/LHCII) of green algae and higher plants, FCP antenna have larger capacity in blue-green light absorption and more diversities in terms of the protein and pigment composition, structural organization and flexibility, which may cope with fluctuating light conditions, enabling diatoms to harvest sufficient energy under dim and rapidly changing light environment under water and to protect the photosynthetic cores from potential radiation damage[7,8].

The representative Lhcb proteins of major LHCII antennae in green plants bind 8 Chls *a*, 6 Chls *b*, 1 neoxanthin, 2 luteins, and 1

[1]Photosynthesis Research Center, Key Laboratory of Photobiology, Institute of Botany, Chinese Academy of Sciences, Beijing, China. [2]University of Chinese Academy of Science, Beijing, China. [3]China National Botanical Garden, Beijing, China. [4]Department of Mechanical Engineering, Tsinghua University, Beijing, China. [5]Research Institute for Interdisciplinary Science, Graduate School of Natural Science and Technology, Okayama University, Okayama, Japan. [6]These authors contributed equally: Songhao Zhao, Lili Shen, Xiaoyi Li, Qiushuang Tao. ✉e-mail: shen@cc.okayama-u.ac.jp; wdwang@ibcas.ac.cn

violaxanthin (a total of 4 carotenoids), which function to harvest blue and red light[9]. In the red branch, Chl *b* is absent whereas Chl *c* is present in some organisms including diatoms. The diatoms have a higher carotenoid/Chl ratio, which, together with the binding of Chl *c*, enable them to harvest more blue-green light underwater. For example, Lhcf4 of *Phaeodactylum tricornutum* contains 7 Chls *a*, 2 Chls *c*, 7 Fxs, and 1 diadinoxanthin (Ddx)[4]. These features give rise to a brown color of the diatoms[10]. The high energy-dissipation capabilities of diatoms are ascribed to the presence of Ddx and diatoxanthin (Dtx), special carotenoids of FCPs[7,8], which offers robust non-photochemical quenching (NPQ) ability to the photosystems of diatoms.

In typical higher plants and green algae, 4–10 monomeric LHCIs are arranged into either one or two semi-spherical belts on one side of the PSI core[11–16], with 2 additional LHCIs associated with the opposite side in green algal PSI-LHCI[14–16], and 6–11 LHCII subunits in both monomeric and trimeric states are connected to PSII core[17–20]. On the other hand, FCPs have distinctly different assembling patterns in both centric and pennate diatoms. In the genome of a centric diatom *Chaetoceros gracilis*, 46 FCP genes are found, which constitute a large antenna pool to support PSI and PSII[21]. Among the FCP antennae of PSII in *C. gracilis*, the dominant FCPIIs are named FCPII-A/B/C according to their molecular weights, and a 22 kDa FCPII-A subunit assembles into tetramers and associates with the PSII core via monomeric FCPII-D and FCPII-E, or by a small subunit PsbG[22–26]. The FCPII-B/C subunits have smaller molecular weights (18–19 kDa), and are suggested to aggregate into trimers like major LHCII trimer[27]. On the other hand, the number of FCPI associated with PSI is much larger, and cryo-electron microscopy (cryo-EM) analyses showed that *C. gracilis* PSI-FCPI (Cg-PSI-FCPI) has 16–24 FCPIs, which are arranged in one closed and one or two semi-closed FCPI rings around the PSI core[28,29]. In another centric diatom *Thalassiosira pseudonana*, PSI-FCPI supercomplex with 18 FCPIs and PSII-FCPII supercomplex with 2 FCPII trimers were proposed to be present in the thylakoid membrane by cryo-electron tomography with a resolution at 20 Å[30]. For FCPII of PSII in diatoms, monomeric, dimeric[4], and tetrameric FCPIIs[24–26] have been found, whereas trimeric FCP resembling the major LHCII proteins of green lineage were not shown ambiguously, although they are also proposed to be present in diatoms[8,30].

In a representative centric diatom *Cyclotella meneghiniana*, two main FCP fractions FCPa and FCPb have been isolated and characterized by biochemical, spectroscopic and electron microscopic approaches[31,32]. FCPa and FCPb are proposed to be trimers and oligomers, respectively, and FCPa comprises of smaller (18 kDa) Lhcf subunit similar to FCPII-B/C, whereas FCPb comprises of larger (19 kDa) Lhcf subunit similar to FCPII-A of *C. gracilis*. However, the detailed structure and their association with either PSI or PSII are unknown. Here, we solved the structures of a PSII-FCPII supercomplex and a trimeric FCP from *C. meneghiniana* by single particle analysis using cryo-EM, which revealed detailed structures of monomeric, dimeric and trimeric FCPs, as well as their binding to the PSII core. The location and arrangement of pigments including Chl *a*, Fx, Ddx and Dtx in FCPs are revealed, providing important insights into the light-harvesting and photoprotection processes in the diatom *C. meneghiniana*.

## Results

### Characters of Cm-PSII-FCPII and trimeric FCP

The Cm-PSII-FCPII and trimeric FCP used for single particle analysis were isolated from *C. meneghiniana* cells cultured at discontinuous, low-intensity light conditions (Methods). A crude Cm-PSII-FCPII sample was obtained by sucrose density gradient ultracentrifugation (SDG) after solubilization of thylakoid membrane with 2.3% α-dodecyl-β-D-maltoside (DDM) (Fig. 1a), followed by gel filtration chromatography (GE, Superose 6 Increase 10/300 GL) (Fig. 1b). Cm-PSI-FCPI cannot be completely separated from Cm-PSII-FCPII by gel filtration, and was co-purified with PSII-FCPII. SDS-PAGE showed that Cm-PSII-

FCPII is the dominating one in the elution peak at 12.8 mL, and a small amount of Cm-PSI-FCPI is present (Fig. 1c), agreeing with the heterogeneous refinement of the classified particles (Supplementary Fig. 1a). We also collected the FCP fractions from the top part of SDG and isolated the trimeric FCPs and mainly dimeric FCPs from them by a further SDG (Fig. 1a). The FCPII peptides dominated in the Cm-PSII-FCPII supercomplex have molecular weights of around 18-19 kDa (Fig. 1c)[27], which is smaller than the tetrameric Cg-FCPII-A[24–26] as well as the isolated FCP dimer and FCP trimer of *C. meneghiniana* (Fig. 1c).

Monomeric Cm-PSI-FCPI, dimeric Cm-PSII-FCPII and trimeric FCP particles were classified from cryo-EM images and processed by the cryoSPARC software (Supplementary Fig. 1a, b). The overall resolutions of the density maps for Cm-PSI-FCPI monomer, Cm-PSII-FCPII dimer and Cm-FCP trimer are 3.32 Å, 2.92 Å, and 2.72 Å respectively (Supplementary Table 1 and Supplementary Fig. 1). We found that more than 60% the dimeric Cm-PSII-FCPII particles lost three peripheral monomers FCPII-I/J/K in one side (Supplementary Fig. 1a), indicating their loose association with the PSII core, which resulted in a lower local resolution for the whole Cm-PSII-FCPII supercomplex (Supplementary Fig. 1c). Based on the analysis of the FCP bands by mass spectrometry and specific sequences obtained via transcriptome sequencing of *C. meneghiniana* and genomic sequence of *C. cryptica*, we assigned three FCP sequences to the local map of FCPII-G/H1/H2 adjacent to CP47/D2/PsbX/PsbH as Cc-Lhcx6_1, Cm-Lhcf7 and Cc-Lhcf7-like protein, respectively (Fig. 2a; Supplementary Fig. 2a–c and Supplementary Fig. 3). FCPII-I is tentatively distinguished as a Lhca2 type protein based on the presence of a long helix-D toward the PSII core in the density map, and a homologous Tp-Lhcf11 sequence was assigned to FCPII-J (Supplementary Fig. 2d, e and Supplementary Fig. 3). Owing to the poor resolution of the FCPII-K map, 13 ligands were docked into its local map but all amino acids were replaced by poly-alanines (Supplementary Fig. 2f).

The trimeric FCP of *C. meneghiniana* was determined as a homotrimer encoded by a gene homologous to *fcp05* of *C. cryptica* (Supplementary Fig. 2g and Supplementary Fig. 3)[32,33]. The molecule weight of Cm-FCP05 is around 19.7 kDa (Fig. 1c), consistent with the previous results[31,32].

### Architecture of the Cm-PSII-FCPII supercomplex

The Cm-PSII-FCPII exists as a dimer with two-fold rotational symmetry and its symmetrical structure resembles the configurations found in Cg-PSII-FCPII and PSII-LHCII from plants (Fig. 2a)[17–20,24–26]. Each PSII core contained 22 subunits (18 transmembrane subunits and 4 extrinsic lumenal subunits), which have high similarities with that of *C. gracilis* (Fig. 2 and Supplementary Fig. 4a). A unique extrinsic subunit of diatom, Psb31[34,35], was missing in Cm-PSII-FCPII (Fig. 2b, c), which is probably released during detergent solubilization, resulting in the low oxygen-evolving activity of the sample (Methods). Another subunit PsbG (Supplementary Fig. 4a) which is responsible for the association of the peripheral tetrameric FCPII-A antennae to the PSII core in Cg-PSII-FCPII[24–26], was also missing in Cm-PSII-FCPII, which may explain the absence of FCP tetramers in this supercomplex.

A major difference between Cm-PSII-FCPII and Cg-PSII-FCPII lies in the peripheral FCPs surrounding the PSII core (Supplementary Fig. 4a). Cm-PSII-FCPII core contains 6 FCPs, among which three monomeric subunits named FCPII-I/J/K are attached to the CP43 side, whereas an FCP heterodimer (FCPII-H1/H2) and a monomeric FCPII-G were linked to the CP47 side (Fig. 2a). Among them, FCPII-I has almost a same position and orientation as that of Cg-FCPII-D (Supplementary Fig. 4a, b). In contrast, the positions of other FCPII monomers and dimer only partially overlap with those of the moderately bound tetramer (MT) and strongly bound tetramer (ST) in Cg-PSII-FCPII (Supplementary Fig. 4a).

No FCP trimer is present in the PSII-FCPII supercomplex observed in the current structure. As free FCP trimers were observed in SDG

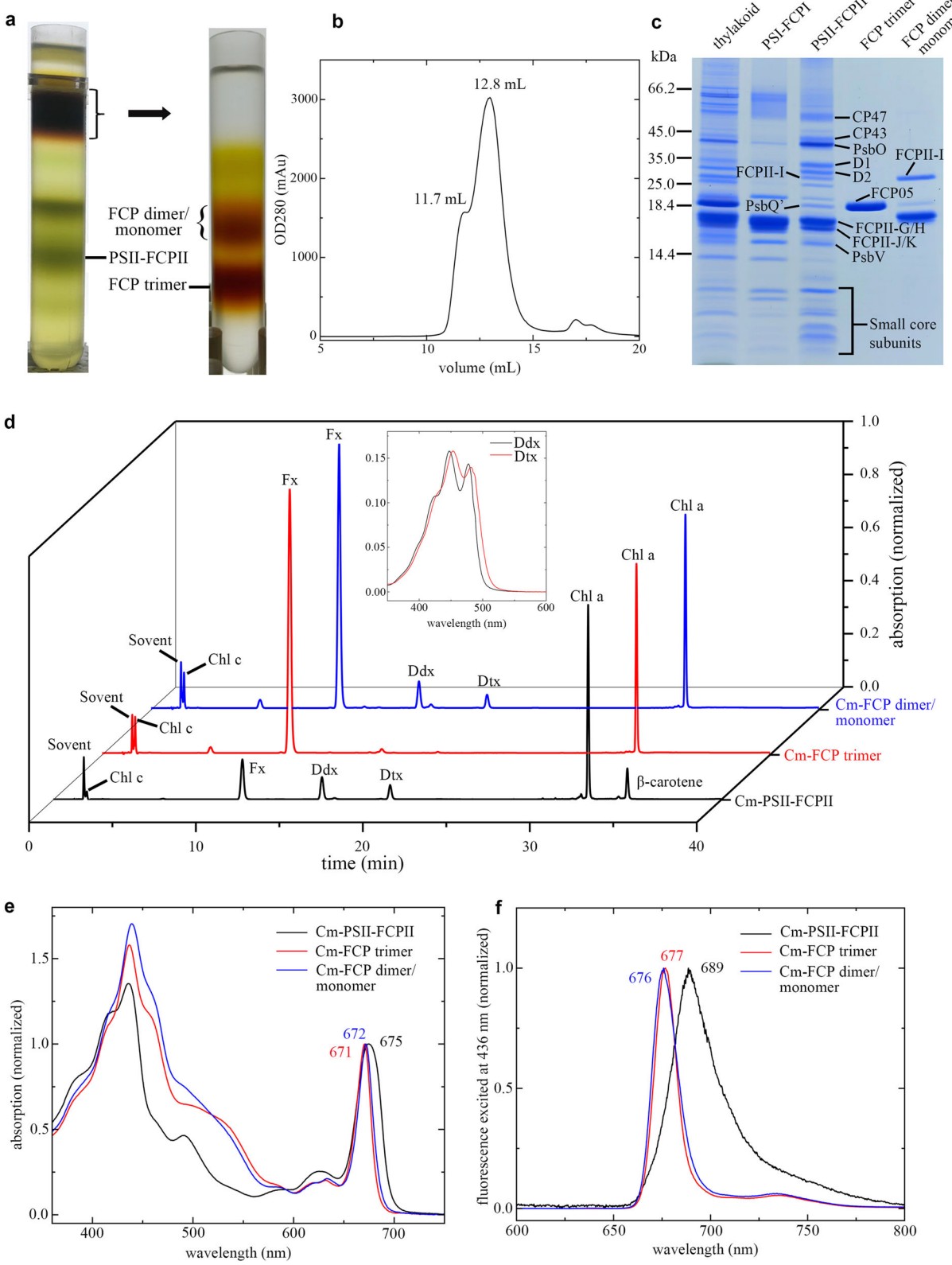

(Fig. 1a), they must be detached from PSII or are originally not associated with PSII. We identified a homotrimeric FCP and some other dimeric FCPs in the free FCP fractions; they are thus not strongly associated with PSII (Fig. 1a, c). The tightly associated FCPII-H dimer and FCPII-I/J/K monomers may bridge the association of these trimeric and dimeric FCPs to the PSII core in *C. meneghiniana*.

In the Cm-PSII-FCPII supercomplex structure, we assigned 206 Chls *a*, 24 Fxs, 2 Dtxs, 12 Ddxs, 20 β-carotenes (BCRs) (Supplementary Table 2). In each subunit of the Cm-FCP trimer, we assigned 8 Chls *a*, 3 Chls *c*, 7 Fxs and 1 sulfoquinovosyl diacylglycerol (SQDG) molecule (Fig. 3a, b and Supplementary Table 2). More Chls *a* and fewer Chls *c* were observed to bind to Cm-FCPII-G/H1/H2 in the present structure,

**Fig. 1 | Isolation and characterization of Cm-PSII-FCPII and Cm-FCP trimer samples. a** Separation of the Cm-PSII-FCPII dimer and Cm-FCPII trimer by sucrose density gradient centrifugation (SDG). Crude Cm-PSII-FCPII dimer was labelled in the first SDG (left tube), and the free FCP complexes on the top were separated in the second SDG (right tube). **b** Further purification of the Cm-PSII-FCPII dimer by gel-filtration. An elution peak at 12.8 ml and a shoulder at 11.7 ml were labelled. **c** SDS-PAGE characterization of related samples. The lanes labelled as PSI-FCPI, PSII-FCPII, FCP trimer and FCP dimer/monomer (termed as dimer in the main text)

correspond to the 11.7 ml and 12.8 ml samples in **b**, FCP trimer and FCP dimer/monomer in **a**, respectively. The main peptides of Cm-PSII-FCPII and Cm-FCP were labeled according to mass spectra analyses. **d** Pigment analysis of Cm-PSII-FCPII dimer, Cm-FCP trimer and Cm-FCP dimer/monomer by HPLC. **e** Absorption spectra of Cm-PSII-FCPII dimer, FCP trimer and FCP dimer/monomer. **f** 77 K fluorescence spectra (excited at 436 nm) of Cm-PSII-FCPII dimer, FCP trimer and FCP dimer/monomer. Each experiment was repeated at least three times independently and all showed similar results.

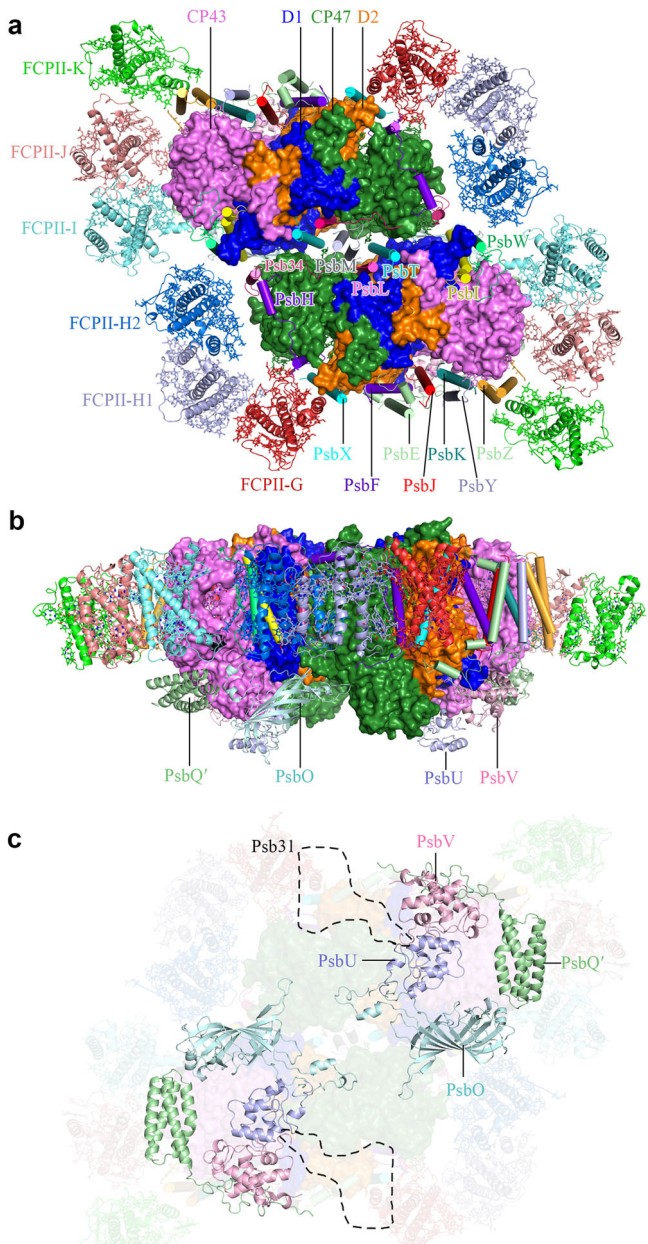

**Fig. 2 | Overall structure of the Cm-PSII-FCPII dimer. a** View of dimeric Cm-PSII-FCPII normal to the membrane plane from the stromal side. **b** View parallel to the membrane plane. **c** View normal to the membrane plane from the lumenal side, where the area circled by a dashed line indicate the absent Psb31 subunit.

in agreement with the pigment analysis (Fig. 1d and Supplementary Table 2). However, a higher content of Chl *c* was found in the FCP dimer and trimer, respectively (Fig. 1d), revealing large differences among the FCP antennae and also suggesting that most Chls *c* of *C. meneghiniana* are associated with the released FCP trimer and dimers.

These results suggest that, although *C. meneghiniana* and *C. gracilis* are both classified as centric diatom species, different strategies are adopted for their FCP assembly within the PSII-FCPII supercomplex, which supports the notion that the diversity of PSII-FCPII in diatoms is higher than that of PSII-LHCII in green algae and higher plants[14–20].

## Structures of peripheral FCPIIs and their interactions around PSII core

As stated above, Cm-FCPII-G was assigned as Lhcx6_1 based on fitting of the sequence with the map (Supplementary Figs. 2a and 3). This Lhcx type subunit has similar N-terminal loop and three trans-membrane helices A/B/C as those of FCPII-A of *C. gracilis* (Cg-Lhcf1) and Lhcf4 of *P. tricornutum* (Pt-Lhcf4) (Supplementary Fig. 5a). The B-C loop of FCPII-G is short and shifted towards its helix-A on the lumenal surface (Supplementary Fig. 5b), leaving a large space to accommodate PsbX (Fig. 2a). Helix C of FCPII-G interacts closely with PsbX (Fig. 4a, b) by a strong hydrogen bond and extensive hydrophobic interactions. In contrast, no FCP antenna is connected to PsbX in the Cg-PSII-FCPII supercomplex, and Cg-FCPII-A tetramer binds to the PSII core through an additional Cg-PsbG subunit (Supplementary Fig. 4a). In analogy to a bridging role of Cg-FCPII-E between the FCPII-A tetramer and PSII core, Cm-FCPII-G has a similar helix D in the C-terminal region (Supplementary Fig. 5c, d). However, a different C-terminus was observed in Cm-FCPII-G, which forms hydrophobic interactions with the C-terminal loop of FCPII-H1 on the lumenal side and offers a binding site for Chl *a*410 (Fig. 4a, c).

We assigned 11 Chls *a* and 3 carotenoid sites in Cm-FCPII-G (Fig. 3c and Supplementary Table 2). Owing to the different loop structures, a relocation of Chl $a405_{FCPII-G}$ from lumenal to the stromal side and a small shift of Chl $a411_{FCPII-G}$ are found (Supplementary Fig. 5e, f), which can potentially link the energy exchange between two stromal Chl clusters (Fig. 5), and further deliver excitation energy efficiently to the PSII core (Fig. 6a, b). This may compensate for the loss of Chl $a400_{Cg-FCPII-E}$ in Cm-FCPII-G. In addition, an extra Chl $a410_{FCPII-G}$ site is found (Supplementary Fig. 5f), which may establish an energy transfer pathway with Chl $a412_{FCPII-H1}$ on the lumenal layer to facilitate energy transfer from FCPII-H1 to FCPII-G (Fig. 6c).

Three carotenoids identified in Cm-FCPII-G are Fx301/Dtx303/Fx305 (Fig. 3c and Supplementary Fig. 2l). The location of $Dtx303_{FCPII-G}$ resembles that of Ddx303 in Cm-FCPII-H and Cg-FCPII-D, which intersects with the pivot cluster Chl $a402$-$a403$-$a406$ and provides a possible NPQ site in the FCPII monomer close to the PSII core (Figs. 3c and 7a, c).

Cm-FCPII-H1/H2 was identified as a heterodimer, despite of 80% sequence identity between them (Supplementary Fig. 6). Cm-FCPII-H1/H2 have typical N-terminal loop and three trans-membrane helices A/B/C as those of Pt-Lhcf4 and Cg-FCPII-A (Supplementary Fig. 5g–i), indicating that they belong to the Lhcf family (Supplementary Fig. 3b). The stromal C-A loop of FCPII-H1/H2 is shorter and protrudes outwards, which accommodates the Chl $a410_{FCPII-H}$ site (Fig. 3d and Supplementary Fig. 5j). A longer C-terminal loop is present in FCPII-H1/H2, offering the docking site for Chl $a412_{FCPII-H}$ (Supplementary Fig. 5j). In addition, the exclusive presence of Chl $a413$ near the lumenal end of helix-B in FCPII-H2 confirms the heterogeneity between FCPII-H1/H2 (Fig. 3d

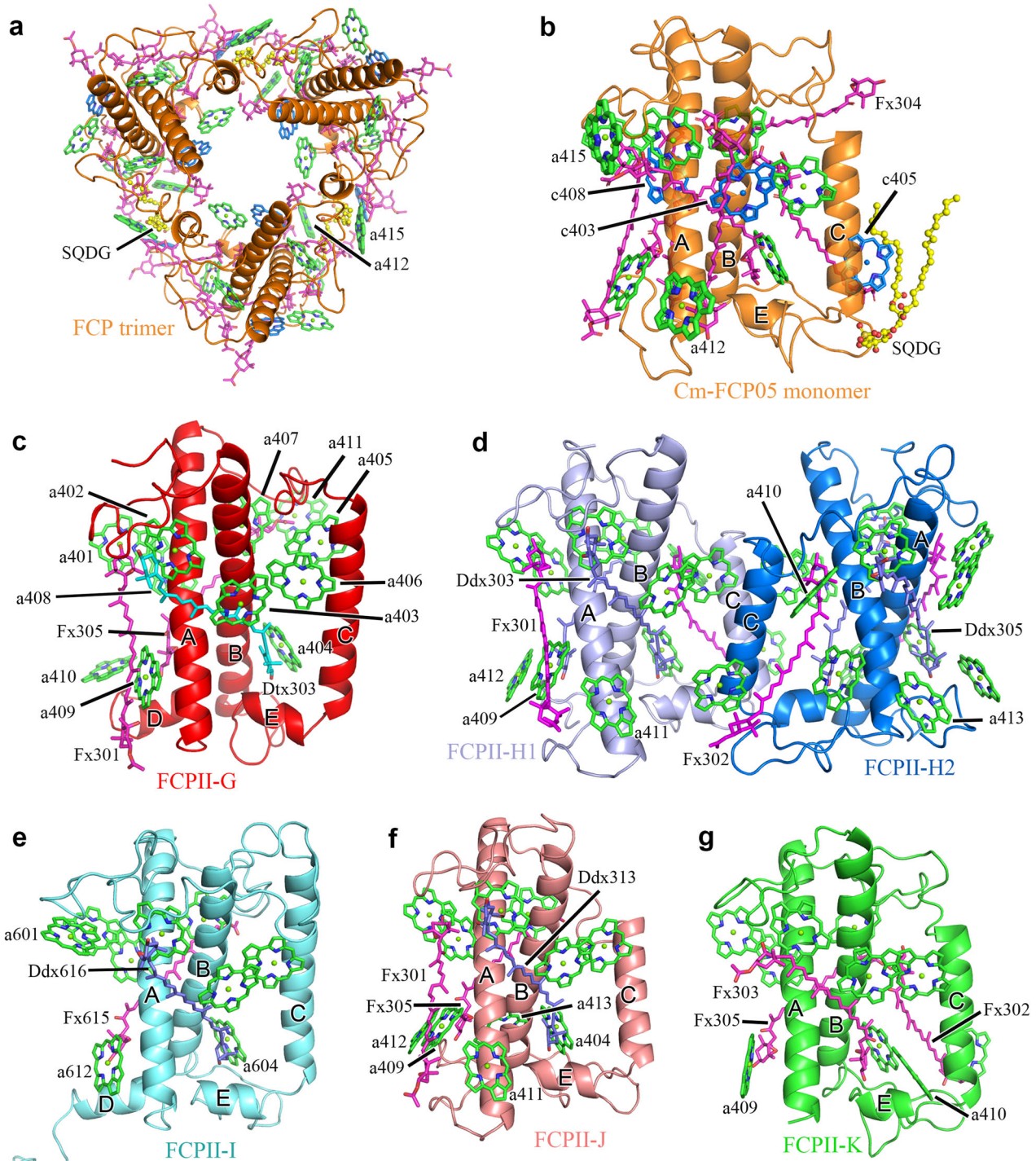

**Fig. 3 | Structures of six peripheral FCPs around the *C. meneghiniana* PSII core and of the FCP trimer.** **a** Structure of the trimeric FCP. **b** Structure of one Cm-FCP05 monomer in the FCP trimer. **c**–**g** Structures and arrangement of pigments, in FCPII-G, dimeric FCPII-H1/H2, FCPII-I, FCPII-J, FCPII-K, respectively. Chl *a*, Chl *c*, Fx, Ddx and Dtx are colored green, marine, purple, blue and cyan, respectively.

and Supplementary Fig. 5i). This Chl *a*413 also establishes an additional excitation energy transfer (EET) pathway with Chl *a*607$_{CP47}$ in the lumenal layer (Fig. 5 and Fig. 6d).

FCPII-H1/H2 dimer is assembled in a "head-to-head" manner by two adjacent helices C, which resembles the Pt-Lhcf4 dimer (Fig. 4a, d and Supplementary Fig. 4c, d). Dimerization of these two FCPII-H subunits can be enhanced by hydrogen bonds (Arg133-Chl *a*406 and Trp120-Chl *a*406) and extensive hydrophobic interactions formed by the helices C and Chls *a*406 (Fig. 4d). The contacts between two helices

C of FCPII-H are strengthened by interactions between their helices E, making two helices C closer on the lumenal side and farther on the stromal side (Supplementary Fig. 4d). Two corresponding Chls *a*406 of FCPII-H1/H2 are inserted between their helixes C and situated closer to each other, shortening their Mg distance from 14.7 Å of Pt-Lhcf4 to 10.7 Å on the stromal side (Supplementary Fig. 4e). Therefore, the two pairs of hydrogen bonds found in the Pt-Lhcf4 dimer are not conserved in FCPII-H on the stromal surface. Comparing with Pt-Lhcf4, Cm-FCPII-H1/H2 shifted toward each other, resulting in a tighter assembling form

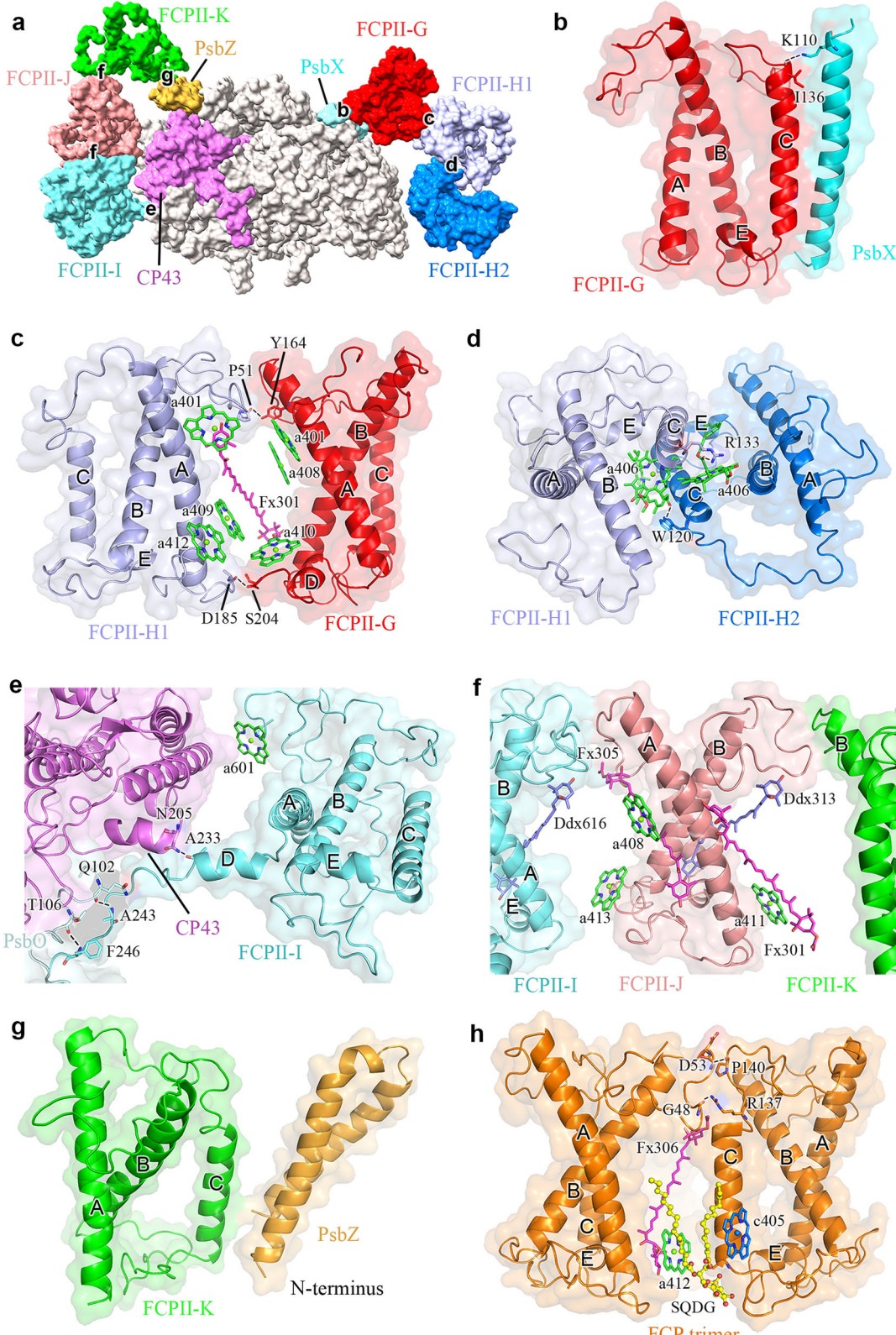

**Fig. 4 | Subunit-subunit interactions within the Cm-PSII-FCPII dimer and FCP trimer. a** Overall view of crucial interactions in a Cm-PSII-FCPII monomer. **b** Interactions between FCPII-G and PsbX. **c** Interactions between FCPII-G and FCPII-H1. **d** Interactions between FCPII-H1 and FCPII-H2. **e** Interactions between FCPII-I and CP43. **f** Interactions between FCPII-J and FCPII-I/K. **g** Interactions between helix C of FCPII-K and C-terminus of PsbZ. **h** Interactions between monomers of an FCP trimer. The crucial hydrogen bonds are indicated by dashed lines, and the Chls, Fxs and lipid located at the interfaces are shown.

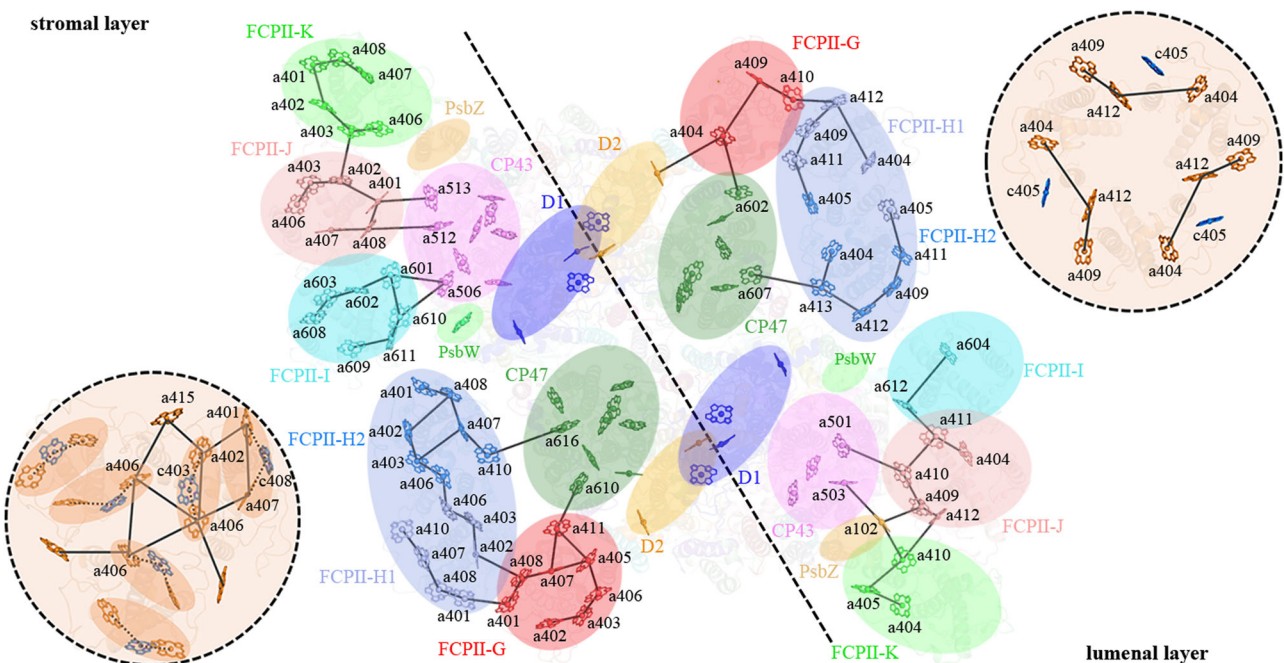

**Fig. 5 | Energy transfer pathways in a Cm-PSII-FCPII monomer and the Cm-FCP05 trimer.** Distributions of Chls and possible energy transfer pathways in a Cm-PSII-FCPII monomer and the Cm-FCP05 trimer are depicted for the stromal layer (left panel) and lumenal layer (right panel), respectively. Solid lines indicate pathways between two adjacent Chls, dashed lines divide two layers in a Cm-PSII-FCPII monomer, and dotted lines indicate Chls *a-c-a* clusters in the Cm-FCP05 trimer.

of the FCP dimer (Supplementary Fig. 4c). As a result, two Chl *a*402-*a*403-*a*406 clusters of the FCP-H1/H2 dimer may form a larger Chl cluster, facilitating the energy transfer between the two FCP monomers (Supplementary Fig. 4e). These differences in the structures and pigment arrangements between the Pt-Lhcf4 and Cm-FCPII-H dimers indicate that diversifications occurred in the assembling pattern, energy harvesting and transfer of FCPIIs surrounding the PSII core of diatoms.

The binding of the FCPII-H dimer to the PSII core is dominated by the association of FCPII-H1 with FCPII-G in their loop regions with help of several crucial residues and pigments (Fig. 2a and Fig. 4a, c). The direct connection between FCPII-H1/H2 and the PSII core is weak because of the missing PsbG (Supplementary Fig. 4a). Location of the additional Chl *a*413 in FCPII-H2 may provide some hydrophobic interactions with the Psb34 subunit to make FCPII-H2 attach to the PSII core (Fig. 6d).

Four carotenoids are assigned in each of FCPII-H1 and FCPII-H2, they are Fx 301/302 and Ddx 303/305 (Figs. 3d and 7). The intersecting Ddx 303/305 are close to the Chl *a*402-*a*403-*a*406 and *a*401-*a*407-*a*408 clusters, respectively, and hence may mediate efficient NPQ under strong light conditions (Fig. 7a, b).

FCPII-I is a monomeric antenna featured with extended helix D on the lumenal side (Supplementary Fig. 4b), which helps to identify the specific amino acid sequences during its model building. The extended helix D interacts with CP43 and PsbO (Fig. 4a, e), serving as one of the major factors to attach FCPII-I to the PSII core, which resembles the role of FCPII-D in *C. gracilis* (Supplementary Fig. 4a). The associations between FCPII-J and the PSII core are weak and indirectly through loop-loop interactions between FCPII-I/K (Figs. 2a and 4a, f). FCPII-K is tightly attached to PsbZ via its helix C (Fig. 4a, g), which resembles the case of Cg-FCPII-E (Supplementary Fig. 4a). Most of the Chls bound to FCPII-I/J/K are similar to that of Cg-FCPII-D and Pt-Lhcf4. Some extra Chls are found in the interfaces between FCPII-I/J/K and the PSII core (Supplementary Fig. 5k). Chl *a*413$_{FCPII-J}$ and *a*411$_{FCPII-J}$ are found in the interface between FCPII-J and FCPII-I and between FCPII-J and FCPII-K, respectively. Correspondingly, the energy transfer among FCPII-I/J/K and from them to the PSII core may be enhanced at the lumenal layer.

The absorption spectra (Fig. 1e) showed that Cm-PSII-FCPII has distinct differences with Cg-PSII-FCPII-A at the blue-green and green light regions[24]. Fewer Chls *c* and Fx are bound in the Cm-FCPII-G/H/I/J/K subunits than those in the Cm-FCP trimer, suggesting that the main FCP antennae responsible for green light harvesting, such as Fcp02, Fcp05 and Fcp06 of *C. meneghiniana*[32], may be loosely attached in the periphery of Cm-FCPII-G/H/I/J/K and have been lost during purification.

### Structure of the trimeric FCP
As seen in Fig. 1a, c, trimeric and dimeric FCP were found as the main FCP antennae on the top of SDG, which are dissociated from the photosystems, agreeing with previous native-PAGE assay[27]. The secondary structure of Cm-FCP05 is similar to that of Pt-Lhcf4 and Cg-FCPII-A except for the existence of a longer C-terminal loop (Fig. 8a, b). However, this Cm-FCP05 is assembled into a "head-to-tail" trimer similar to that of the plant LHCII trimer (Figs. 3a and 8c, d), which is largely different from the previously reported Pt-Lhcf4 dimer and Cg-FCPII-A tetramer[4,24–26]. On the other hand, we found that the structure of three Cm-FCP05 monomers did not match well with those of the plant LHCII trimer, which is reflected by a deviation of a monomer at around 29.3° viewed from the stromal side (Fig. 8d). This assembly pattern is due to the differences found in the loop structures between Cm-FCP05 and plant LHCII, which induces different monomer-monomer interactions at the stromal side (Figs. 4h and 8f). Two hydrogen bonds between N-terminal loop and the C-A loop (Fig. 8f) make helix B of one monomer closer to helix C of the adjacent monomer, leaving large distances among three helices B and a hole at the stromal side (Fig. 8e). On the other hand, the C-terminal region of Cm-FCP05 is shorter than LHCII and the helix D is missing (Fig. 8c), making interactions between adjacent FCP05 monomers indirect at the lumenal side and the appearance of specific connections via several ligands (Fig. 8g). An extra Chl *a*412 and an SQDG molecule are inserted into the gap between helices A and C at the lumenal surface, building hydrophobic interactions and hydrogen bonds with Chls *c*405, Fx306 and Thr118 of B-C loop (Fig. 8g). Due to this assembly pattern, the three Cm-FCP05 monomers form two enlarged holes at

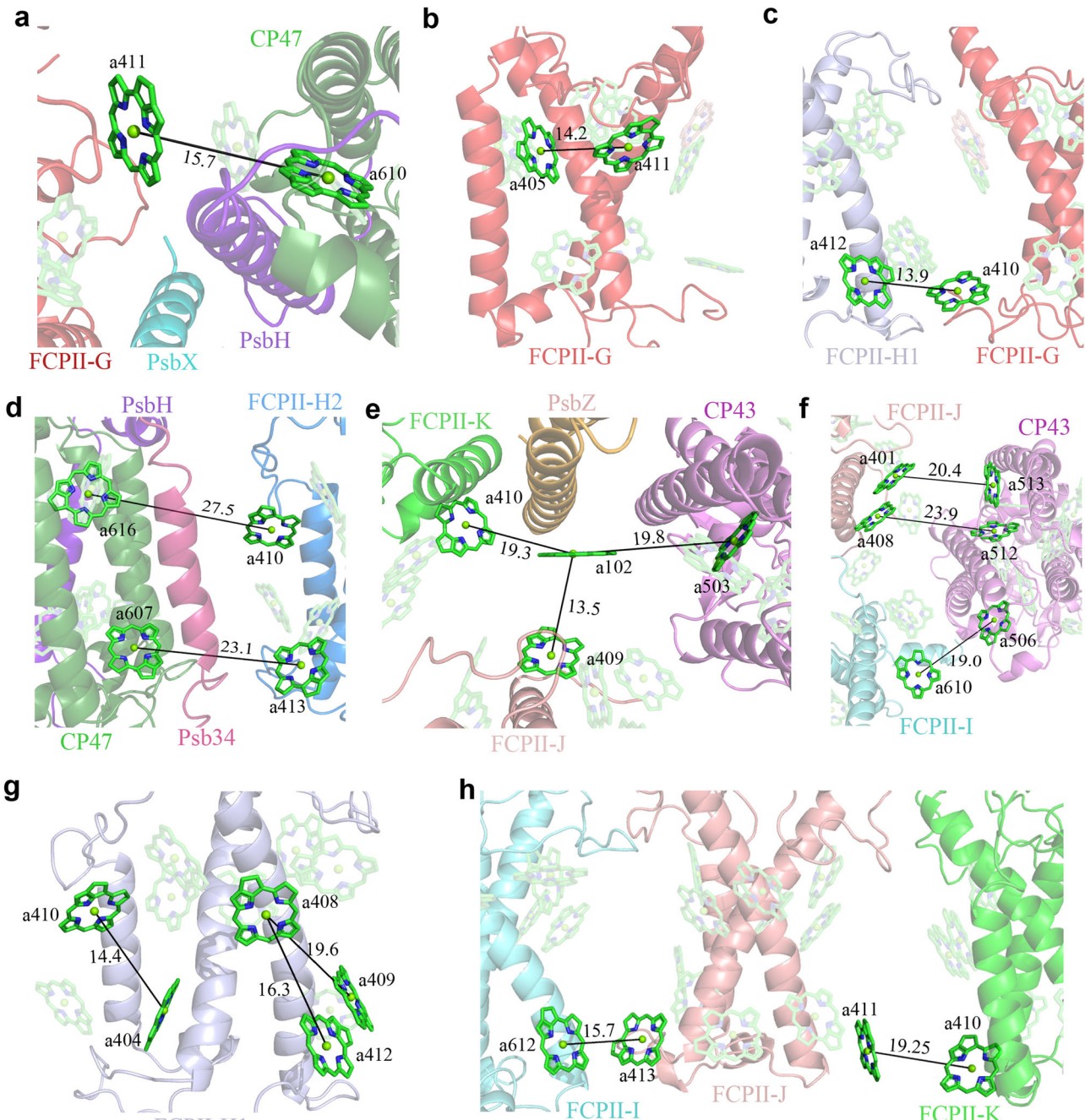

**Fig. 6 | Possible energy transfer pathways inside a Cm-PSII-FCPII monomer.** **a** Energy transfer from FCPII-G to CP47. **b** Energy-transfer within FPCII-G. **c** Lumenal energy transfer from FCPII-H1 to FCPII-G. **d** Stromal/lumenal energy transfer from FCPII-H2 to CP47. **e** Possible lumenal energy transfer from FCPII-J/K to CP43 mediated by Chl *a*102 of PsbZ. **f** Possible stromal energy transfer from FCPII-I/J to CP43. **g** Energy transfer between stromal/lumenal layers within FCPII-H1. **h** Lumenal-layer energy exchange between FCPII-J and FCPII-I/K.

both stromal and lumenal sides to accommodate more ligands, as indicated by increased distances among helices A and B at around 10 Å (Fig. 8e). These ligands include a specific "red Fx304" with red-shifted absorption to harvest more green light[4], which locates horizontally at the stromal surface and enhances green light harvesting of the Cm-FCP05 trimer (Fig. 3b), and two Chls located in the interface regions between the adjacent monomers, Chl *a*412 at the lumenal layer and Chl *a*415 at the stromal layer (Fig. 3b), which are crucial for energy transfer among the monomers (Fig. 5).

Distribution of Chls and energy transfer pathways are affected by the assembly of the Cm-FCP05 trimer. Because no direct interactions between the Cm-FCP05 trimer and the PSII core were observed, we examined the arrangement of Chls in the Cm-FCP trimer at the lumenal and stromal sides, respectively (Fig. 5). At the stromal side, one specific Chl *a*402-*c*403-*a*406 cluster is located in the inner side of the trimer to hold the energy equilibrium among three monomers, and another Chl *a*401-*c*408-*a*407 cluster in the exterior region is responsible for EET to the PSII core or other FCP antennae. The distances among Chls *a*406 in the Cm-FCP05 trimer are larger than those in the Pt-Lhcf dimer; for this reason, an extra Chl *a*415 was found in the Cm-FCP05 trimer which may play a role in enhancing the efficient EET between the adjacent monomers. At the lumenal side, Chl *a*409 and Chl *a*412 have a small

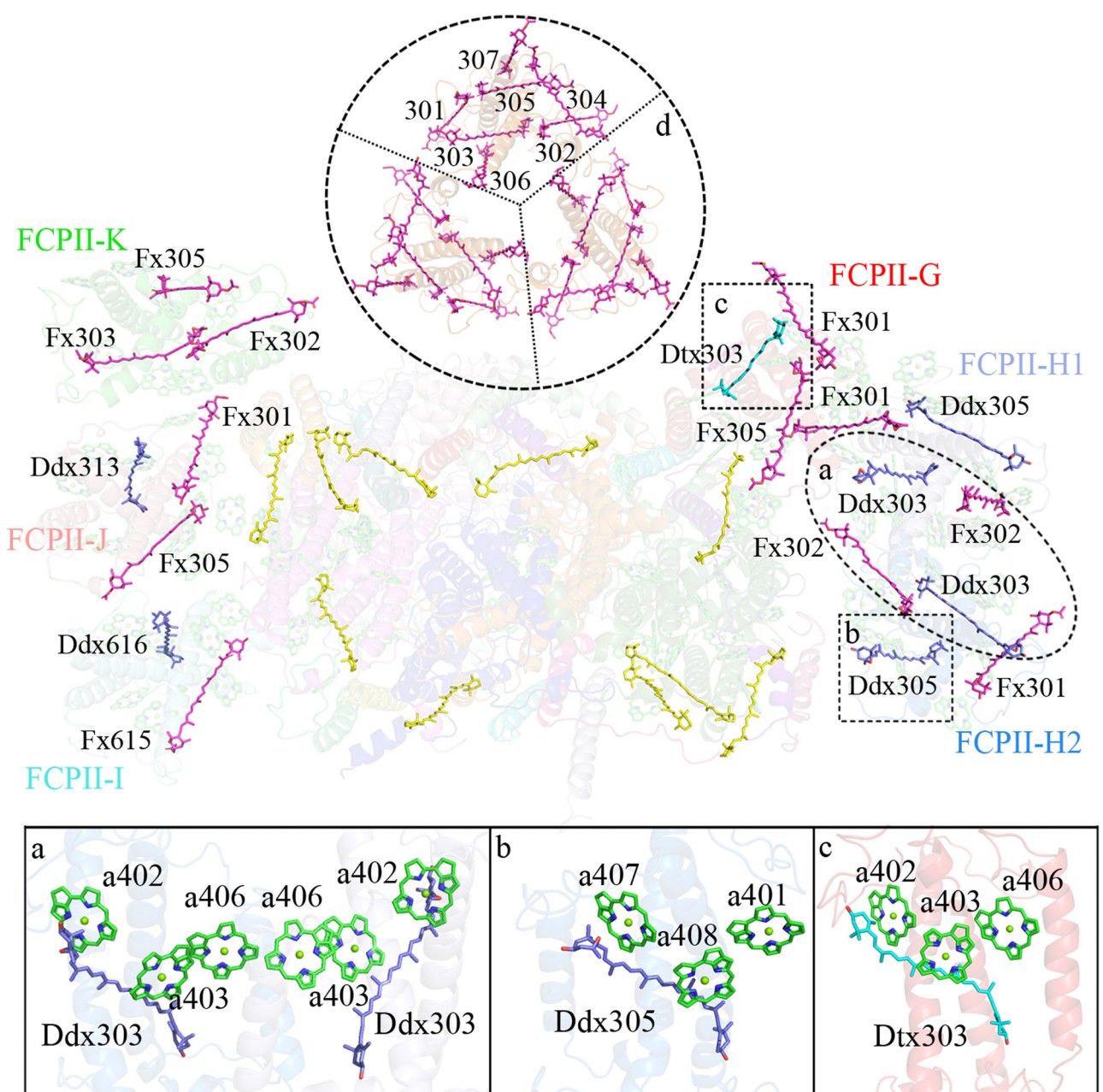

**Fig. 7 | Distribution of carotenoids and potential sites for the Ddx-Dtx cycle in a Cm-PSII-FCPII monomer.** Three areas circled by dashed lines are enlarged in **a**–**c** panels to show Ddx303 in the FCPII-H dimer, Ddx305 in FCPII-H1/H2 and Dtx303 in FCPII-G. **d** Fxs in Cm-FCP05 trimer.

distance, enabling an efficient EET to Chl $a$404 of the adjacent monomer.

**Possible energy transfer pathways in Cm-PSII-FCPII**

The monomeric FCPII-I/J/K adjacent to CP43 may transfer their energy separately to the core. On the lumenal layer, Chl $a$102$_{PsbZ}$ can accept the energy from both Chl $a$409$_{FCPII-J}$ and $a$410$_{FCPII-K}$, which is then transferred to Chl $a$503$_{CP43}$ (Fig. 6e). The role of PsbZ as a relay to mediate the inward energy transfer has been reported in Cg-PSII-FCPII[24–26]. On the stromal layer, Chl $a$610$_{FCPII-I}$ and $a$401/$a$408$_{FCPII-J}$ are linked to Chl $a$506/$a$512/$a$513 of CP43 respectively (Fig. 6f), which partially results from the interactions between helix A of FCPII-J/K and the PSII core. In contrast, the inward energy transfer processes are not distributed homogeneously in the case of FCPII-G/H1/H2. Fast energy-transmitting pathway from the FCP antennae to the core may occur for FCPII-G, which is from Chl $a$411$_{FCPII-G}$ to $a$610$_{CP43}$ (Fig. 6a). Slower

energy-delivering pathways are identified for FCPII-H2, where energy may be transferred from Chl $a$413$_{FCPII-H2}$ to $a$607$_{CP43}$ and from $a$410$_{FCPII-H2}$ to $a$616$_{CP47}$ (Fig. 6d) due to their longer distances. No efficient energy transfer to the core can be discerned for FCPII-H1, indicating that FCPII-H1 tends to deliver its energy to CP47 indirectly via FCPII-G and FCPII-H2 (Fig. 5).

The lumenal and stromal energy transfer pathways based on the corresponding Chl layers are extensively interconnected, rather than independently paralleled. Representative cases can be found in FCPII-H1/H2, where the stromal Chl $a$408 and $a$410 are connected to the lumenal Chl $a$409/$a$412 and $a$404 respectively (Fig. 6g). The efficient energy exchanges between the two layers are crucial, as interfacial energy transfer between subunits relies more on the pathway of a certain layer in some cases, such as stromal and lumenal energy transfer from FCPII-G and FCPII-K to the PSII core (Fig. 6a, e).

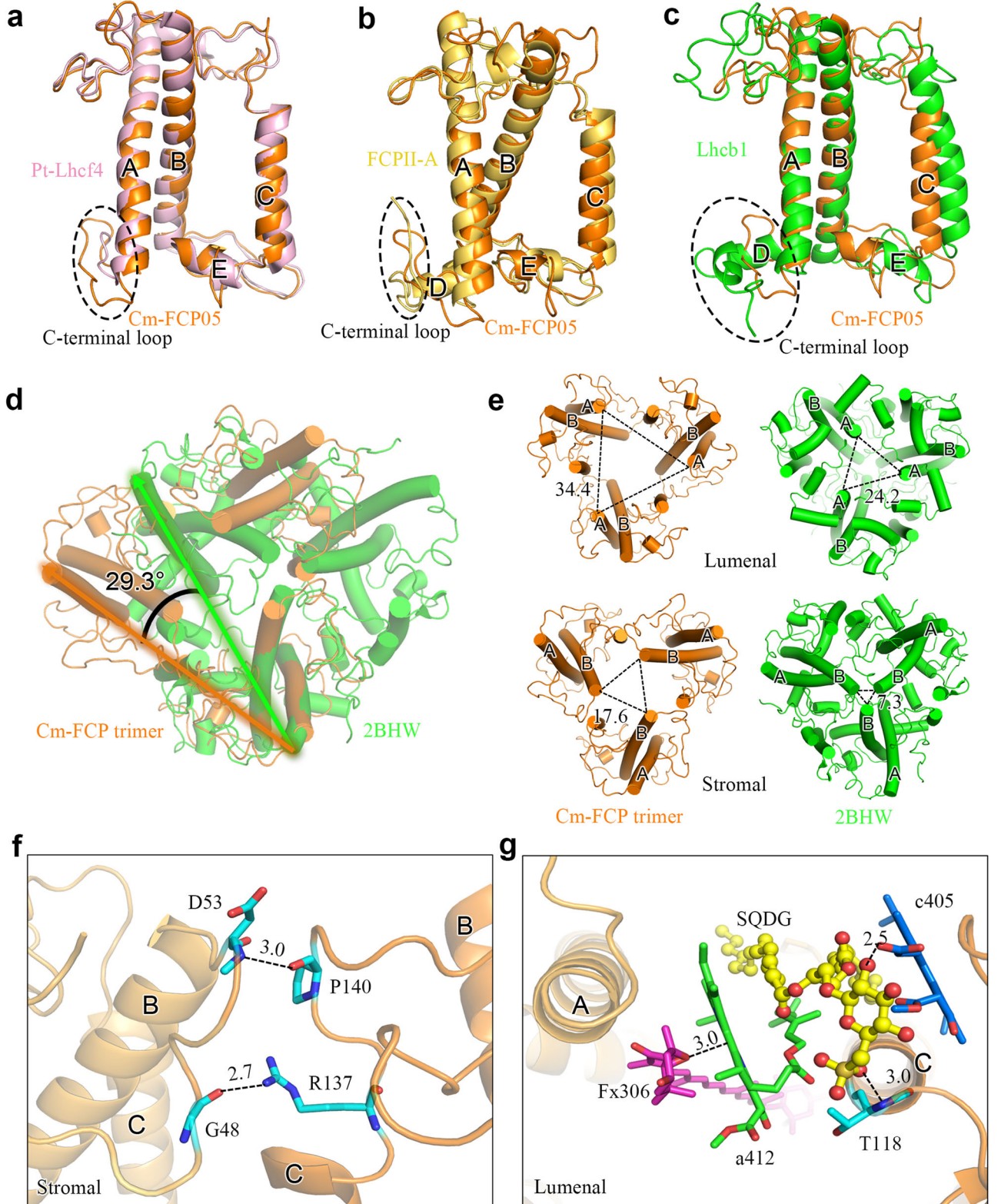

**Fig. 8 | Unique organization of the Cm-FCP trimer. a–b** Comparison of proteins between the Cm-FCP trimer (Cm-Fcp05) with Pt-Lhcf4, FCPII-A, and LHCII mono-mer (Lhcb1). **c** The C-terminal loop associated with the assembly of FCPs was circled by dashed lines. **d** Comparison of the Cm-FCP trimer (orange) with the LHCII trimer (PDB code 2BHW) (green). **e** The distances among three monomers of Cm-FCP-trimer and plant LHCII trimer. The distances among three helices A at the lumenal side and helices B at the stromal side were labelled. **f**, **g** Interactions between monomers of the Cm-FCP trimer at the stromal and lumenal sides, respectively.

Different features of pigment distribution are found in Cm-FCPIIs compared with Cg-FCPII-A and Pt-Lhcf4. Most of the Chl 403 and Chl 408 sites in Cm-PSII-FCPII are occupied by Chl $a$ instead of Chl $c$ (Supplementary Table 2). Consequently, additional energy transfer pathways can be established, which includes the stromal connection of Chl $a610_{FCPII-I}$ with Chl $a506_{CP43}$ (Fig. 6f). The coupling strength of Chls inside the triangular Chl clusters $a402$-$a403$-$a406$ and $a401$-$a407$-$a408$ of Cm-FCPII-G/H is also increased, which may result in a lower energy level, potentially protecting the core from excess energy supply. In particular, the tighter dimerization pattern of FCPII-H1/H2 leads to further coupling of two Chl $a402$-$a403$-$a406$ clusters from FCPII-H1/H2, forming a large Chl cluster with lower energies, which may trap and delay the inward energy transfer from the loosely associated FCPIIs in the periphery of the FCPII-H dimer. Another feature is the assignment of extra Chls like Chl $a410$/$a412$/$a413$, which are not seen in Cg-FCPII-A and Pt-Lhcf4. These extra Chls may act as pivot points for energy transfer. Chl $a411_{FCPII-J}$ and $a413_{FCPII-J}$ are connected to Chl $a410_{FCPII-K}$ and $a612_{FCPII-I}$, respectively, on the lumenal layer, establishing the efficient links between the monomeric FCPII-I/J/K (Fig. 6h). In addition, Chl $a412_{FCPII-H1}$ is linked to Chl $a410_{FCPII-G}$ (Fig. 6c), providing an efficient energy transfer pathway between these two subunits.

### Green light harvesting and photo-adaption associated with carotenoids

Carotenoids are important for light-harvesting and excess energy dissipation via NPQ to protect the reaction center. In contrast to the typical lutein, neoxanthin, violaxanthin and zeaxanthin in LHCs of green algae and higher plants[36], keto-type Fx is found in FCPs which improves the green light harvesting capability, and the Ddx/Dtx cycle is involved in mediating photoprotection[6–8]. In previous FCP structures, various Fx sites were identified and some of them were suggested as green light harvester[4]. The epoxy group of a Ddx was found close to the stromal side, which may facilitate its de-epoxidation to Dtx. An increased amount of Dtx is usually related to high-light treatment and quenching states of FCPs[32,37,38]. Although our *C. meneghiniana* cells were cultured under a low-light condition, considerable amount of Dtxs are present in the isolated Cm-PSII-FCPII sample in addition to the Ddxs as shown in the high-performance liquid chromatography (HPLC) analysis (Fig. 1f). In the structure of Cm-PSII-FCPII, 6 Fxs, 4 Ddxs and 1 Dtx are identified in the Cm-FCPII-G/H region, whereas only 2-3 carotenoids are tentatively assigned in each antenna due to the lower local resolution of FCPII-I/J/K (Fig. 7). Possibilities exist that more Ddx/Dtx may be found or some of the current Fx in FCPII-I/J/K may be replaced by Ddx/Dtx if density maps are improved to a higher resolution.

In the monomeric and dimeric Cm-FCPIIs structures associated with the PSII core, a reduced amount of Fxs was found. Three red Fxs 304/306/307, which may increase green light absorption in the Pt-Lhcf4 dimer and Cm-FCP05 trimer structures, are lacking in the Cm-FCPIIs, suggesting that the Cm-PSII-FCPII has a weaker ability to collect green light compared with the Cg-PSII-FCPII supercomplex[24], in consistent with their absorption features shown in Fig. 1e. The main trimeric and dimeric Cm-FCP on the top of SDG contains more red Fxs and are probably loosely bound outside of FCPII-G/H/I/J/K in vivo, and are responsible for green light harvesting under deep water.

One Dtx is identified at the 303 site of FCPII-G in the current structure, which shares the same location and orientation as Ddx303 in Cg-FCPII-D (Fig. 7c). Strong NPQ capacity may be established here, since Dtx303 in this Lhcx6_1 type Cm-FCPII-G is accompanied by the Chl $a402$-$a403$-$a406$ cluster (Fig. 7c), which may serve pivot roles in the energy transfer pathway. Previous ultrafast spectroscopic analysis has predicted the existence of one NPQ site firmly bound to the diatom PSII[37,38]. The present structure may suggest that Dtx303 corresponds to this site, which is situated on the CP47 side and has direct contacts with the PSII core. The transient NPQ component generated by initial

Dtxs may therefore be formed here, which is predicted by ultrafast spectroscopic analysis of the cells[39]. Additionally, 2 Ddxs assigned to the 303 and 305 sites of FCPII-H1/H2 suggest that FCPII-H plays different roles compared with the Pt-Lhcf4 dimer, Cm-FCP05 trimer and Cg-FCPII-A tetramer (Fig. 6a, b and Fig. 7d). Thus FCPII-G and FCPII-H are both inner FCP antennae strongly associated with the PSII core and probably mediate EET from outside FCPs, and may also play photoprotection roles through the Ddx-Dtx cycle under high light conditions.

## Discussion

The cryo-EM structures of the PSII-FCPII supercomplex and the disassociated FCP trimer (FCP05) of *C. meneghiniana* revealed distinctive features in the organization of FCPII around the PSII core and the FCP05 monomer comprising of the trimer. Six FCPII subunits are found to associate with the PSII core, which includes three monomeric FCPIIs, an FCPII heterodimer and an FCP subunit Lhcx6_1 possibly involved in photoprotection. This number is much less than 11 FCPII subunits found in PSII-FCPII from *C. gracilis*[24–26]. Furthermore, two FCPII tetramers are found in Cg-PSII-FCPII, whereas no tetramer is found and only a dimer is present in Cm-PSII-FCPII. These results indicate a great diversity in the organization of FCPII around the PSII core, which may reflect the adaptation of different diatoms to their living environment, mainly the light intensities and their fluctuations.

In the periphery of Cm-PSII-FCPII, the Lhcx6_1 type Cm-FCPII-G directly binds to PSII and mediates interactions between the dimeric Cm-FCPII-H1/H2 and the PSII core. This binding mode is apparently different from the previously found Cg-FCPII-A tetramers in Cg-PSII-FCPII[24–26]. No tetrameric FCP was found in the free FCP pool of *C. meneghiniana*, which suggests a species divergence between the two species of diatoms, *C. gracilis* and *C. meneghiniana*. Although the crystal structure of dimeric FCP has been elucidated in the pennate diatom *P. tricornutum*[4], the Pt-PSII-FCPII complex remains unsolved. The current structure of Cm-PSII-FCPII provides important clues on how FCPII dimer is associated with the PSII core and functions in energy harvesting and excess energy dissipation.

The structure of the Cm-FCPII-H1/H2 dimer exhibits similarities to that of Pt-Lhcf4 dimer[4], as both form a C2 symmetry through their helices C. However, Cm-FCPII-H adopts a tighter assembly for the formation of FCP dimers, which is attributed to different Lhcf sequences involved. Due to the protein and structural variations, Cm-FCPII-H1/H2 binds fewer Fx molecules compared to the Pt-Lhcf4 dimer, resulting in a weak absorption in the green light region. Nevertheless, it binds more Ddx molecules and all of its Chls are identified as Chls $a$, indicating that Cm-FCPII-H1/H2 may serve primarily as an energy transfer relay from peripheral FCPs to the PSII core.

The Cm-FCPII-H1/H2 dimer demonstrates strong interactions with the PSII core facilitated by Lhcx6_1 (FCPII-G), which exhibits similar pigment composition to Cm-FCPII-H1/H2. The presence of a significant amount of Chls $a$ without Chls $c$ in Lhcx6_1 enhances the efficiency of energy transfer. Additionally, the identification of a Dtx303 site in Lhcx6_1 suggests its involvement in photoprotection through the deepoxidation of Ddx, which is consistent with the presence of a photoprotection site near the diatom PSII core[32,40–43]. The pigment network and associated energy transfer in Cm-PSII-FCPII indicate that the location of Lhcx6_1 may serve as a critical site between Cm-FCPII-H1/H2 and the PSII core for energy collection and potential energy quenching. This Chl $a$ and Ddx/Dtx abundant feature of FCPII-G/H1/H2 implies their potential involvement in the NPQ process, which may correspond to the qE1 site close to the PSII core in diatoms[33,37].

The Cm-FCP05 trimer found here may have weaker associations with the PSII core and detach into free FCPs, which are recovered on the top of SDG. The Cm-FCP trimer has more Fxs and Chls $c$, which may play a crucial role in capturing blue-green light similar to that of the Pt-Lhcf4 dimer[4]. Cm-FCP05 exhibits a unique "head-to-tail" assembly

pattern due to its specific loops and lipids, forming enlarged spaces between each monomer to accommodate more Chls and carotenoids. In addition, a red Fx304 molecule positioned horizontally on the stromal surface is found, which is similar to that of Pt-Lhcf dimer[4]. The trimeric Cm-FCP05 structure revealed in this study represents the atomic structure of FCP trimer containing a substantial amount of red Fx and Chls c, enabling the capture of blue-green light[40]. This trimer is not observed directly in the isolated Cm-PSII-FCPII supercomplex, which suggests that it is released during the detergent solubilization. This in turn suggests that it is bound to the outer region of PSII and contributes to light-harvesting and energy transfer under weak light. When the light intensity is increased, detachment of the Cm-FCP05 trimer from PSII core or formation of aggregates may occur to quench the excess excitation energy in different ultrafast time scales, which may be realized via singlet and triplet states between coupled Chl clusters and adjacent Fxs to avoid photodamage[39–41,44,45].

In conclusion, the cryo-EM structures of PSII-FCPII and the trimeric FCP of *C. meneghiniana* provide detailed insights into the structures of various FCP antennae and the assembly of FCPs within the PSII supercomplex. Thus, FCP monomer, dimer, trimer and tetramer have been found in different species of diatoms, reflecting a great diversity in the organization of FCP subunits, which may be a result of adaptation to the light environment that diatoms experience and/or evolution. The different organization of Cm-PSII-FCPII and Cg-PSII-FCPII, and their differences with those of PSII-LHCII from the green lineage organisms, provide a deeper understanding of light harvesting and utilization in diatoms. These findings will shed light on the adaptation mechanism of diatoms as successful photosynthetic organisms in the modern ocean.

## Methods

### Isolation and purification of the Cm-PSII-FCPII supercomplex and Cm-FCP trimer

The strain *Cyclotella meneghiniana* (CCMA-274) was obtained from the Center for Collections of Marine Algae at Xiamen, China. Cells were cultured in an artificial seawater medium bubbled with 3% $CO_2$ at 20 °C under discontinuous low-intensity light (14 h light and 8 h dark alternatively; 40 μmol photon $m^{-2}s^{-1}$). The optical density of cell culture finally reached to 0.8 at 750 nm before harvesting. All following procedures were performed under dim green light and at 4 °C or on ice. Cells were harvested by centrifugation at 5000 × g and resuspended in an MMB buffer (30 mM 2-morpholinoethanesulfonic acid (Mes), 5 mM $MgCl_2$, 1 M betaine, pH 6.5). Resuspended cells were disrupted by a high-pressure homogenizer AH-D150 (ATS ENGINEERING LIMITED, CHINA) with 5 cycles operated at 300 Bar. Subsequent centrifugation at 2000 × g for 10 min is conducted to remove unbroken cells. The remaining thylakoids in the supernatant were pelleted by centrifugation at 150,000 × g for 1 h by a discontinuous sucrose gradient centrifugation (0.5/1.3 M sucrose in an MNC buffer containing 30 mM Mes, 10 mM NaCl, 5 mM $CaCl_2$, (pH 6.5). The resulted fraction between 0.5-1.3 M sucrose was collected and resuspended in an MBNC buffer (30 mM Mes, 1 M betaine, 10 mM NaCl, 5 mM $CaCl_2$, pH 6.5) at 1.5 mg Chl/mL, which was stored at -80 °C for further treatment.

The thylakoid membranes are solubilized at 1.0 mg Chl/mL by 2.3% dodecyl-α-D-maltopyranoside (α-DDM) for 20 min with stirring at 300 rpm. The mixture was centrifuged at 40,000 × g for 20 min to remove the insolubilized debris. The remaining supernatant was uploaded onto a continuous SDG prepared by freeze-thawing of the MNC buffer supplemented with 0.65 M sucrose and 0.03% α-DDM. Centrifugation was conducted at 200,000 × g for 20 h. The crude Cm-PSII-FCPII fraction (Fig. 1a) was collected by syringe and further concentrated by 100-kDa cutoff concentrator, which was then subjected to a gel filtration chromatography (GE, Superose 6 Increase 10/300 GL) with the MNC buffer containing 0.03% α-DDM and

100 mM NaCl as the mobile phase (Fig. 1b). Fractions peaked at 12.8 mL were collected and concentrated with a 100-kDa cutoff membrane filter up to 3.0 mg Chl/mL for single particle analysis.

The FCP fraction from the top part of SDG was collected, and trimeric and dimeric FCPs were isolated from them by a further SDG which was prepared by freeze-thawing of the MNC buffer supplemented with 0.4 M sucrose and 0.03% α-DDM. Cm-FCP trimer was collected by a syringe and concentrated up to 1.0 mg Chl/mL for single particle analysis by a 50-kDa cutoff membrane filter.

### Characterization of the Cm-PSII-FCPII supercomplex and Cm-FCP trimer

Transcriptome sequencing was performed to seek FCP sequences of *C. meneghiniana* as reported previously[24], which resulted in 9 FCP-related sequences. Subunit compositions of the sample were analyzed by sodium dodecyl sulfate-polyacrylamide gel electrophoresis with a gel containing 16% polyacrylamide and 7.5 M urea[46]. The FCPs bands were cut out and subjected to mass spectrometry (MS) analysis as described previously[47] (Fig. 1c), whose results were searched against the 9 *C. meneghiniana* sequences obtained via transcriptome sequencing and the fcp01-12 sequences reported previously[32,48,49]. The Chl concentration was determined as previously reported[50], which shows that the ratio of Chl a/c is as high as 100. Pigment compositions were analyzed by HPLC analysis as reported previously[51], which identified the existence of Chl a, Chl c, Fx, Ddx, Dtx and BCR based on the corresponding absorption spectra and elution times (Fig. 1d). In particular, Ddx and Dtx were distinguished according to the absorption spectra reported[52]. Absorption spectrum at room temperature (Fig. 1e), fluorescence emission spectrum at 77 K and oxygen-evolving activity at 25 °C are measured according to the methods described for Cg-PSII-FCPII[24]. The fluorescence spectrum excited at 436 nm was peaked at 689 nm (Fig. 1f). The oxygen-evolving activity of *C. meneghiniana* thylakoid and Cm-PSII-FCPII are 237.9 and 74.1 μmol $O_2$ $(mg Chl)^{-1} h^{-1}$. Loss of the oxygen-evolving activity of Cm-PSII-FCPII may be caused by the high concentration of the detergent used for the solubilization.

The absorption spectrum at room temperature, fluorescence emission spectrum at 77 K of the Cm-FCP trimer, and pigment compositions of the trimeric and dimeric FCPs, were measured with the same method as for Cm-PSII-FCPII.

### Cryo-EM data collection

An aliquot of 2.5 μL of the purified Cm-PSII-FCPII sample at a concentration of 2.0 mg Chl/mL was applied to holey carbon grids (Quantifoil R2/1, Au, 300 mesh; glow-discharged for 60 s in the air). The grid was blotted for 2 s with a blotting-force of -1 after waiting for 2 s, which was then plunged into liquid ethane using Vitrobot under the condition of 100% humidity and 8 °C. A total of 10941 super-resolution images were captured with an FEI 300 kV Titan Krios electron microscope and a Gatan K3 summit direct electron detector at a magnification of 22,500×, with a defocus range from -1.5 to -2.5 μm, yielding a nominal pixel size of 1.06 Å. Each exposure of 5.6 s was fractioned into 32 movie frames, resulting in a total dose of ~50 e/Å².

The purified Cm-FCP trimer at 1.0 mg Chl/mL was applied to a glow-discharged holey carbon grid (CryoMatrix Amorphous alloy film R1.2/1.3, 300 mesh) and vitrified using a Vitrobot (FEI) at 100% humidity and 4 °C. The sample was blotted for 4 s with a blotting-force of 2. Cryo-EM images were collected on a Titan Krios microscope operated at 300 kV equipped with a Gatan Quantum energy filter, with a slit width of 20 eV, a K3 camera (Gatan) operated at the super-resolution mode, with a magnification of 81,000×. Each stack of 32 frames was exposed for 1.8 s, with a total dose of ~60 e/Å². A defocus range of -1.0 to -2.0 μm was used on the EPU software (Thermo Fisher Scientific). The final images were binned, resulting in a pixel size of 1.04 Å for further data processing.

## Data processing

A total of 7153 movies were corrected by motion correction and CTF correction using cryoSPARC[53], from which 1,192,896 particles were picked by crYOLO[54] and used for 2D classification in cryoSPARC. After several rounds of 2D classification, 979,148 high-quality particles were selected for hetero refinement. Two of the classes after hetero refinement (769,405 particles related to PSII-FCPII) were selected for 3D classification focusing on FCPII-I/J/K, resulting in 157,205 complete PSII-FCPII complex particles for the subsequent non-uniform refinement using C2 symmetry (Supplementary Fig. 1a). A part of PSI particles were also found in the results of hetero refinement (88,200 particles). After 3D classification for the PSI particles, 66,522 particles were selected for non-uniform refinement using C1 symmetry (Supplementary Fig. 1a). The overall resolutions of PSII-FCPII and PSI-FCPI are 2.92 Å and 3.32 Å based on the gold-standard criteria, FSC = 0.143[55] (Supplementary Fig. 1c, d, f, g). We took advantage of the C2 symmetry of PSII to double the number of peripheral FCPII particles, and used focused FCPII-I/J/K and FCPII-G/H regions to design soft masks for local particle subtraction. As a result, 314,409 particles were selected for local refinement of FCPII-G/H and FCPII-I/J/K. Finally, the local FCPII-G/H map was refined at 3.23 Å resolution and the FCPII-I/J/K map was refined at 3.48 Å resolution based on the gold-standard FSC with a cutoff value of 0.143 using local refinement in cryoSPARC (Supplementary Fig. 1c)[53].

The density map of Cm-PSI-FCPI fits well with the core structure of Cg-PSI-FCPI including the specific PsaS subunit on the stromal side, consistent with its solid association with PsaC/D/E subunits to stabilize the electron transfer components (Supplementary Fig. 7a, b). In total, 13 FCPI antennae arranged in the whole innermost layer (FCPI-1/2/3/4/5/6/7/8/9/10/11) and two subunits of the middle layer (FCPI-12/13) are found in the Cm-PSI-FCPI structure (Supplementary Fig. 7c), whereas other FCPIs found in *C. gracilis* are missing, either due to their release during solubilization by the high concentration of detergent or their original absence in this diatom. The orientations and locations of the innermost-layer FCPIs, as well as the pigments bound, of Cm-PSI-FCPI, are almost the same as those of Cg-PSI-FCPI, except for the slight shift of FCPI-12/13 (Supplementary Fig. 7c), indicating that these two subunits are rather flexible.

A total of 3953 movies of Cm-FCP trimer were processed with cryoSPARC[53] and 1,471,158 particles were boxed using crYOLO[54]. The particles chosen were extracted from micrographs, and several rounds of 2D classification were performed to select particles of high quality, which resulted in 970,425 particles for the subsequent non-uniform refinement using C3 symmetry in cryoSPARC (Supplementary Fig. 1b). This results in a density map with an overall resolution of 2.72 Å based on the gold-standard criteria, FSC = 0.143 (Supplementary Fig. 1e, h).

## Model building and refinement

For model-building of Cm-PSII-FCPII, the PSII core of Cg-PSII-FCPII (PDB code 6JLU) was docked into the density map of Cm-PSII-FCPII after removing the subunits Psb31 and PsbG in Chimera[56], which fits well with the density map. The sequences for FCPII-G and FCPII-H can be identified by the local density maps with the help of the mass spectrometry analysis results. FCPII-G was fitted with the sequence of Cc-Lhcx6_1, and FCPII-H1/H2 heterodimer was assigned as Lhcf7 of *C. meneghiniana* and Lhcf7-like protein of *C. cryptica* (Supplementary Figs. 3 and 6). The Cc-Lhca2 sequence equipped with a long helix-D domain was assigned to FCPII-I according to the peculiar density map for its helix-D region. We also assigned a homologous Cc-Lhcf11 to FCPII-J. Initial models for FCPII-G/H/I/J were generated by homolog modelling using SWISS-MODEL server[57]. Sequence of FCPII-K cannot be determined due to its poor density map, and thus this subunit was assigned as poly-alanines. Adjustment of the backbone, side chains and ligands were performed manually with COOT[58]. The models of the whole Cm-PSII-FCPII and local FCP regions were refined in real space against the density map by Phenix[59]. Geometric problems like atomic clashes were manually edited in COOT, which was refined again in Phenix. Iterations of these two steps were conducted to obtain the final model. The Chls and carotenoids assigned in the current structure are summarized in Supplementary Tables 2 and 3. Chls *a* and *c* are distinguished by the phytol chain, whereas Fx, Dtx and Ddx are assigned mainly by the two end groups. It is worth noting that the pigment sites 408 in FCPII-H and some sites in FCPII-I/J/K may potentially be assigned as Chls *c*. However, due to the limited resolution of the map, we temporarily assigned them as Chls *a* now.

For the model-building of Cm-PSI-FCPI, the PSI core of Cg-PSI-FCPI (PDB code 6LY5) was docked into the density map of the core. FCPI-01 ~ 13 of Cg-PSI-FCPI was fitted into the region of the antennae one by one. Adjustments of the backbones, side chains and cofactors were then performed manually with COOT[58]. The atomic structure of the Cm-FCP trimer was predicted by SWISS-MODEL using the Pt-Lhcf4 model and then refined in real space using COOT and Phenix. Structures in this manuscript were displayed with UCSF ChimeraX[60] and PyMOL[61].

## Reporting summary

Further information on research design is available in the Nature Portfolio Reporting Summary linked to this article.

## Data availability

The cryo-EM maps of the Cm-PSII-FCPII supercomplex and Cm-FCP trimer have been deposited in the Electron Microscopy Data Bank with accession codes EMD-35987 and EMD-36054. The structural models of Cm-PSII-FCPII and Cm-FCP trimer were deposited in the Protein Data Bank under PDB ID 8J5K and 8J7Z. The local maps and corresponding models of Cm-FCPII-G/H and Cm-FCPII-I/J/K are deposited with accession codes EMD-37267/PDB 8W4O and EMD-37268/PDB 8W4P, respectively.

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

## Acknowledgements

The cryo-EM data were collected at the Center for Biological Imaging (CBI), Core Facilities for Protein Science at the Institute of Biophysics, Chinese Academy of Sciences. We thank J.D., P.X., P.H., M.L., and L.Z. for the preparation of the grids; X.L., L.C., and X.H. for the collection of data; X.M. for mass spectrometric analysis; Y.Y. for HPLC analysis; N.G. for the help in the reconstruction of the density map. The project is funded by the National Key R&D Program of China (2021YFA1300403, 2019YFA0906300), the Strategic Priority Research Program of CAS (XDA26050402, XDB17000000), the Youth Innovation Promotion Association of CAS (2020081), the CAS Interdisciplinary Innovation Team (JCTD-2020-06), the CAS Project for Young Scientists in Basic Research (YSBR-093), the National Natural Science Foundation of China (32222007, 31970260, 32270260, 32070267), the Innovation Platform for Academicians of Hainan Province (2022YSCXTD0005), and the Science & Technology Specific Project in Agricultural High-tech Industrial Demonstration Area of the Yellow River Delta (2022SZX12).

## Author contributions

W.W. and J.-R.S. conceived and coordinated the project. Q.T. and S.Z. prepared and characterized the Cm-PSII-FCPII sample. S.Z., Q.T. and X.L. prepared the cryo-EM grids. S.Z., L.S., W.W. and X.L. collected and processed the images. S.Z., L.S., and W.W. built and refined the structural models. S.Z. and L.S. performed the computational analysis of energy transfer. L.S. prepared and characterized the Cm-FCP trimer sample. L.S. and Z.L. collected and processed the images. L.S. and Z.L. built and refined the structural model. C.X., C.Z., Y.Y., M.S., G.H., L.-J., and T.K. contributed to the discussions and comments on the experimental results. The manuscript was written and modified by S.Z., W.W., J.-R.S., L.S., Q.T., and X.L.

## Competing interests

The authors declare no competing interests.
