## [Peer Review File · Nature Communications]

Structural insights into photosystem II supercomplex and trimeric FCP antennae of a centric diatom *Cyclotella meneghiniana*Reviewers' Comments:

Reviewer #1:

Remarks to the Author:

The manuscript by Zhao et al describes the structure of the PSII-FCP complex from a centric strain of diatoms. This is an interesting addition to the already determined structures from *C. gracilis*, the novelty of the current work is in discovering the new form of the trimeric FCP which resembles the green lineage LHCII and in providing the structural basis for energy transfer into the PSII core from the surrounding FCP's in *C. meneghiniana*. From that perspective the manuscript merits publication in N.comm.

My main point of criticism has to do with the data processing and modeling of work which MUST to be substantially improved. The authors acknowledge that the local resolution on three FCP's is very low, but they did not take the essential steps to try and remedy the situation. In addition, their modeling is not done, despite providing satisfactory overall statistics, there are gross mistakes in the modeling even in places where map density is sufficient. I've provided a few examples below which I was able to find within 20 minutes of studying the model.

Map densities around FCP-J,I, K must be improved using focused refinement and classification in region which I'm sure will also improve the overall resolution of the map. As it stands, image processing should be regarded as partially complete.

If this approach does not improve map resolution around FCP-J,I,K then their modeling should reflect that, as partial chains, modeled as poly alanine. I do appreciate the way the authors address these shortcomings in the text, which clarify the limitations of the model to the reader, however these limitations in map resolution should also be clear in the model itself (or preferably improved according to my suggestion above).

I have included a few additional points below.

The manuscript includes many references to NPQ with regards to chl-car configuration, while this is correct, triplet quenching is just as important and should be discussed as well.

Map sharpening and the use of any aberration correction are not described.

Figure 4 can be integrated into other figures.

Line 94 "comprises a" – comprises of

Line 207 – "compensate for the miss" – I suggest "compensate for the loss"

Chain B/b 498-504 not covered by map.

Cm-FCPII-G- supplementary figure 6b shows that appropriate secondary structure restraints were not used during refinement and includes an additional cyan colored helix which is not described in the legend (probably PsbX). I'm not sure the ellipse highlighting the BC loop is correctly placed.

Chain 8 (FCPII-K) - 104-119 – is not covered by map.

Chain 1 – n-terminus is built into very noisy density.

Chain p should be modeled more conservatively. Probably without most side chains.

Chain 4 – some pigments are not supported by the map, most side chains as well.

Chl 413 chain 2 is misplaced (rotated).

Chain 3, this type of discrepancy between map and model is unacceptable:

Detail from chain 1, showing a problematic residue.

Reviewer #2:

Remarks to the Author:

The manuscript by Zhao et al. describes cryo-EM structures of a PSII-FCPII supercomplex and a FCP trimer from a centric diatom *C. meneghiniana*. Diatoms are a major group of algae, greatly contributing to the primary production in the ocean. Although several structures of diatom PSII-FCPII, PSI-FCPI as well as the dimeric FCP were previously determined, the present work reports the trimeric organization of free FCP, as well as the PSII core assembled with monomeric and dimeric FCPs, thus providing novel information of the assembly pattern of diatom FCP in both PSII-free and PSII-bound state.

The cryo-EM structures are of high resolution, and the experimental data are solid. The results are novel, and will be of great interest in the field of photosynthesis. However, the writing and presentation of this manuscript could be greatly improved.

General comments:

1. The manuscript is a bit difficult to read for non-experts. The description of the structure seems somewhat dry. Many detailed descriptions can be shortened. For example, the section "Structures of peripheral FCPIIs and their interactions around PSII core" could be condensed.
2. Several paragraphs and sentences need to be rewritten due to the unclear meaning. For example, Lines 190-193, 231-233, 403-404, etc. English needs to be improved.
3. Figures should be numbered in the order they are cited in the manuscript.

Specific points:

4. Fig. 1a. Is there any evidence that the upper band in the right tube is "FCP dimer"? It could be FCP monomer, since the monomeric FCP-I/J/K are all loosely bound to the core.
5. Fig. 1c. Please label the major bands in the PSII-FCPII, FCP trimer and FCP dimer samples. It would be also interesting to know the identity of the band around 25 kDa in FCP dimer sample.
6. Line 114. Resembles the structures of Cg-PSII-FCPII and PSII-LHCII from plants?
7. Is FCP-H dimer a homodimer? Please specify.
8. Line 150-153. Please clarify whether *C. meneghiniana* does not contain the PsbG gene, or the PsbG protein is present in but detaches from the supercomplex.
9. Line 174-175. It is evident that the three HPLC curves in Fig. 1f were normalized to the Chl a content. However, PSII-FCPII contains much more Chl a than FCP trimer and dimer due to the large amount of Chl a bound to the core subunits. Therefore, a lower peak of Chl c in PSII-FCPII sample does not necessarily mean that it binds very few Chl c.
10. Line 234-235. It is assumed that Fx, Ddx and Dtx are highly similar in their structures. According to Fig. S1f, the resolution of the FCP-H dimer region is around 3.5 angstrom, can the authors distinguish different carotenoids in this region?
11. Line 314. C-terminal -> N-terminal
12. Line 325. What is "a red Fx304"?
13. Line 338-339. According to Fig. 5, Chls 406 and 412 are located at opposite sides of the thylakoid membrane.

Revisions made in response to the comments by Reviewer #1

Reviewer #1 (Remarks to the Author):

The manuscript by Zhao et al describes the structure of the PSII-FCP complex from a centric strain of diatoms. This is an interesting addition to the already determined structures from *C. gracilis*, the novelty of the current work is in discovering the new form of the trimeric FCP which resembles the green lineage LHCII and in providing the structural basis for energy transfer into the PSII core from the surrounding FCP's in *C. meneghiniana*. From that perspective the manuscript merits publication in *N.comm*.

Author's answers:

We greatly appreciate your positive comments and insightful questions on our manuscript. We have improved the maps and models, and revised the manuscript based on your suggestions. We provided point-by-point responses to your questions in the following. All responses are highlighted in red in this text, and all revisions in the original text are recorded by the "track changes" function or highlighted in yellow.

My main point of criticism has to do with the data processing and modeling of work which MUST be substantially improved. The authors acknowledge that the local resolution on three FCP's is very low, but they did not take the essential steps to try and remedy the situation. In addition, their modeling is not done, despite providing satisfactory overall statistics, there are gross mistakes in the modeling even in places where map density is sufficient. I've provided a few examples below which I was able to find within 20 minutes of studying the model.

Map densities around FCP-J, I, K must be improved using focused refinement and classification in relation which I'm sure will also improve the overall resolution of the map. As it stands, image processing should be regarded as partially complete.

Author's answers:

Thank you very much for your valuable suggestions, and we sincerely apologize for not providing a detailed local density map for the FCPII-I/J/K and FCPII-G/H. We admit that despite a global resolution of the PSII-FCPII map at 2.93 Å, the previous local maps for the peripheral FCPII-I/J/K subunits had a poor resolution at around 5-6 Å. As a result, we had to rely on previously published FCP crystal structure to determine the protein backbone and some pigment positions of the FCPII-I/J/K subunits.

Now, following your suggestions we have re-processed the cryo-EM images. We have strategically devised a soft mask focusing on FCPII-I/J/K during 3D classification to get more good particles and remove dirty particles. Furthermore, we selected 157,205 particles with a high-quality for final refinement at the 2.92Å resolution. The improvement over the previous 2.93 Å resolution is limited, but peripheral antennae can be more clearly characterized.

In addition, we took advantage of the C2 symmetry of PSII to double the number of FCPII particles and focused the FCPII-I/J/K and FCPII-G/H regions to design new soft masks for particle subtraction. As a result, 314,409 particles were selected for local refinement. As shown in

Supplementary Figure 1, we have improved the resolution of the local map of the FCPII-I/J/K region to 3.48 Å and the FCPII-G/H region to 3.23 Å. We subsequently refined the models for FCPII-G/H and FCPII-I/J/K at the enhanced resolution, which we believe are much better than the previous ones. We have deposited the revised local maps and corresponding models in the PDB database (codes 8W4O and 8W4P).

If this approach does not improve map resolution around FCP-J, I, K then their modeling should reflect that, as partial chains, modeled as poly alanine. I do appreciate the way the authors address these shortcomings in the text, which clarify the limitations of the model to the reader, however these limitations in map resolution should also be clear in the model itself (or preferably improved according to my suggestion above).

Author's answers:

The image processing and model building have been improved based on your suggestions as described above. Based on the present maps and models, we identified FCPII-J as a Cc-Lhcf11 type protein which is shown in the figure below. However, it is still difficult to identify the outermost FCPII-K antenna due to the poorer local map quality. Following your advice, we have substituted the protein main chain with poly-alanines. This aspect is further explained in the "Method: Model Building" section.

I have included a few additional points below.

The manuscript includes many references to NPQ with regards to chl-car configuration, while this is correct, triplet quenching is just as important and should be discussed as well.

Author's answers:

Thanks for your suggestion, we used 2 triplet quenching references (Agostini et al. 2021; Di Valentin et al. 2012) and added two sentences in the discussion section as follows.

This Chl *a* and Ddx/Dtx abundant feature of FCPII-G/H1/H2 implies their potential involvement in the NPQ process, which may correspond to the qE1 site close to the PSII core in diatoms. (lines 455-457, revised manuscript).

When the light intensity is increased, detachment of the Cm-FCP05 trimer from PSII core or formation of aggregates may occur to quench the excess excitation energy in different ultrafast time

scales, which may be realized via singlet and triplet states between coupled Chl clusters and adjacent Fxs to avoid photodamage. (lines 471-474, revised manuscript).

Map sharpening and the use of any aberration correction are not described.

Author's answers:

Thanks for your comment. The maps uploaded to the database were not subjected to sharpening and aberration correction. In fact, we optimized per-particle defocus and per-group CTF parameters for the final non-uniform refinement, as detailed in the "Data Processing" section of the revised manuscript.

Figure 4 can be integrated into other figures.

Author's answers:

Thank you for your suggestion. We think Figure 4 is an important and comprehensive figure to illustrate the interactions between peripheral FCP and the core subunits. It is also challenging to integrate 7 panels into other figures. Thus, we kept Figure 4 in the text, but optimized and reorganized Figure 4 to show the whole description and the relevant details more clearly.

Line 94 “comprises a” – comprises of

Author's answers:

Thanks for your suggestion. It is corrected in the revised text.

Line 207 – “compensate for the miss” – I suggest “compensate for the loss”

Author's answers:

Thanks for your suggestion. The phrase was changed as suggested.

Chain B/b 498-504 not covered by map.

Author's answers:

Thank you very much for your suggestion. This was indeed an oversight on our part when building the model. The density for chain B/b 483-504 is almost not visible in the map, so we have removed this C-terminal segment in the latest version of the PDB as shown in the figure below.

Previous global map

Current global map

Cm-FCPII-G- supplementary figure 6b shows that appropriate secondary structure restrains were not used during refinement and includes an additional cyan colored helix which is not described in the legend (probably PsbX). I'm not sure the ellipse highlighting the BC loop is correctly placed.

Author's answers:

Thank you for your suggestion. We have made the corrections to the secondary structure depicted in new Supplementary Figure 5b and other related pictures. The cyan PsbX has been labeled and described in the legend.

In the previous Supplementary Figure 6b, the ellipse only circled the region where PsbX conflicts with BC loop of Pt-Lhcf4. In the new version, we highlighted both BC loops of Cm-FCPII-G and Pt-Lhcf4 separately to indicate the shift of Pt-Lhcf4 loop in Cm-FCPII-G. We also added one sentence in the legend to illustrate that significant changes of the BC loop of Cm-FCPII-G is important for its binding to PsbX.

Chain 8 (FCPII-J) - 104-119 - is not covered by map.

Author's answers:

Thank you for your comment. Due to the limited resolution of our previous map, we encountered challenges in accurately aligning the loop region's sequence with the map. We addressed this problem by performing local refinement of the FCPII-I/J/K regions and subsequently modified the models based on the improved local map. However, the resolution of the map for chain 8 (FCPII-J) remains limited. Therefore, we removed many amino acid side chains in this region. Currently, the PDB representation of chain 8 (FCPII-J) - 104-119 in the map is displayed as shown in the figure below, which we believe is much improved than the previous map.

Previous global map

Current global map

Current local map

Chain 1 - n-terminus is built into very noisy density.

Author's answers:

We sincerely apologize for our poor model based on the previous bad local map. We made significant improvements by revising this segment based on the new local density map. The model for the N-terminus of chain 1 in the latest PDB is now assigned correctly, as shown in the figure below.

Chain p should be modeled more conservatively. Probably without most side chains.

Author's answers:

Similar to the questions above, we assigned most of the amino acid residues in the helical region of chain p according to the improved local map, and some side chains of amino acid residues in the loop region for which the map is not visible were removed. For example, side chains in the C-terminal region were identified based on the new local map as shown in the figure below.

Chain 4 - some pigments are not supported by the map, most side chains as well.

Author's answers:

Thanks for your comment. Indeed, the local density map for chain 4 was relatively poor in the previous global map, making it challenging to assign some pigment sites. We have improved the local map, and based on the improved local map of chain 4 (FCPII-K), we are able to determine the exact positions of each pigment and deleted one Chl site. However, it is still difficult to identify

the pigment types and the amino acid sequence of this subunit, owing to its outermost location and higher flexibility. Therefore, we substituted chain 4 with poly-alanines as displayed in the figure below.

Chl 413 chain 2 is misplaced (rotated).

Author's answers:

Thanks for your suggestion. Based on the improved local map, we corrected the model of Chl 413 chain 2 as shown in the figure below.

Previous global map

Current local map

Chain 3, this type of discrepancy between map and model is unacceptable:

Author's answers:

We apologize for the issue that arose in the previous global map. We have improved the model building for chain 3 under the new global and local maps as shown in the figure below, and the structure is based on the improved local map.

Previous global map

Current global map

Current local map

Detail from chain 1, showing a problematic residue.

Author's answers:

Thank you for pointing out this issue. We modified this chain 1 84TYR site as shown in the figure below.

Previous global map

Current local map

Revisions made in response to the comments by Reviewer #2

Reviewer #2 (Remarks to the Author):

The manuscript by Zhao et al. describes cryo-EM structures of a PSII-FCPII supercomplex and a FCP trimer from a centric diatom *C. meneghiniana*. Diatoms are a major group of algae, greatly contributing to the primary production in the ocean. Although several structures of diatom PSII-FCPII, PSI-FCPI as well as the dimeric FCP were previously determined, the present work reports the trimeric organization of free FCP, as well as the PSII core assembled with monomeric and dimeric FCPs, thus providing novel information of the assembly pattern of diatom FCP in both PSII-free and PSII-bound state.

The cryo-EM structures are of high resolution, and the experimental data are solid. The results are novel, and will be of great interest in the field of photosynthesis. However, the writing and presentation of this manuscript could be greatly improved.

Author's answers:

Thank you very much for your positive comments and valuable suggestions to improve our manuscript. According to the comments and suggestions of the reviewer, we have improved the models, checked the language and revised the manuscript. The modifications we made in response to the reviewer's comments are as follows.

General comments:

1. The manuscript is a bit difficult to read for non-experts. The description of the structure seems somewhat dry. Many detailed descriptions can be shortened. For example, the section “Structures of peripheral FCPIIs and their interactions around PSII core” could be condensed.

Author's answers:

We apologize for using excessive structural details which made the manuscript somewhat difficult to read. According to the reviewer's comments, we reduced around 320 words in the section "Structures of peripheral FCPIIs and their interactions around PSII core", and also made some small changes in other places throughout the text.

2. Several paragraphs and sentences need to be rewritten due to the unclear meaning. For example, Lines 190-193, 231-233, 403-404, etc. English needs to be improved.

Author's answers:

Thanks for your suggestions. We have made revisions to these sentences as follows:

Lines 190-193: Original: “The helix C of FCPII-G interacts closely with PsbX (Fig. 4a) with extensive hydrophobic interactions, which is distinctly different from an additional PsbG subunit in PSII-FCPII-A of *C. gracilis*, and holds the antenna protein strongly with the PSII core.”

Modified: “Helix C of FCPII-G interacts closely with PsbX (Fig. 4a, b) by a strong hydrogen bond and extensive hydrophobic interactions. In contrast, no FCP antenna is connected to PsbX in the

Cg-PSII-FCII supercomplex, and Cg-FCPII-A tetramer binds to the PSII core through an additional Cg-PsbG subunit (Supplementary Fig. 4a)” (lines 203-207, revised manuscript).

Lines 231-233: Original: “Consequently, the energy delivery mediated by Cm-FCP-H and Cm-FCP-G may be more favorable than the Pt-Lhcf4 dimer on the stromal side.”

To avoid too much description of structural details, this sentence and associated part were deleted.

Lines 403-404: Original: “Fewer Fxs are found to occupy the 301, 302, 303 and 305 sites in the present monomeric and dimeric Cm-FCPIIs structure associated with the PSII core.”

Modified: “In the monomeric and dimeric Cm-FCPIIs structures associated with the PSII core, a reduced amount of Fxs was found” (lines 392-393, revised manuscript).

3. Figures should be numbered in the order they are cited in the manuscript.

Author’s answers:

Thanks for your suggestion. We have adjusted the numbering of Figures and their corresponding citing in the manuscript based on the order of the figures.

Specific points:

4. Fig. 1a. Is there any evidence that the upper band in the right tube is “FCP dimer” ? It could be FCP monomer, since the monomeric FCP-I/J/K are all loosely bound to the core.

Author’s answers:

Thanks for your comments. The band at the top right in Figure 1a is mainly composed of FCP dimers, as indicated by the elution peak position on the gel filtration column, which closely matches that of Pt-FCP dimers, as shown in the figure below. However, some monomeric FCPs exists, which was revealed by SDS-PAGE and the peak of gel filtration as below. Therefore, we have revised the upper band in the right tube in Figure 1a to "FCP dimer/monomer," and made corresponding modifications in other images of Figure 1 as well as in the manuscript.

5. Fig. 1c. Please label the major bands in the PSII-FCPII, FCP trimer and FCP dimer samples. It would be also interesting to know the identity of the band around 25 kDa in FCP dimer sample.

Author's answers:

*Thank you for your valuable suggestions. We have now labeled the major bands based on the mass spectrometry results in new Fig. 1c. The FCP trimer represents a purified sample, with its peptide homologous to the *C. cryptica* FCP05 protein (encoded by the *fcp05* gene). The FCP dimer/monomer (previously referred to as FCP dimer) is a mixture of many Cm-FCP proteins, but except for the FCP05 protein. Notably, the band around 25 kDa in this sample was identified as the *Lhca2* protein, which corresponds to the FCPII-I subunit and shares homology with *Cg-FCPII-D*.*

6. Line 114. Resembles the structures of *Cg-PSII-FCPII* and *PSII-LHCII* from plants?

Author's answers:

Thank you for your suggestion. Yes, it is "PSII-LHCII from plants", and clarified all of them have two-fold rotational symmetry of PSII core and peripheral antennae.

The Cm-PSII-FCPII exists as a dimer with two-fold rotational symmetry and its symmetrical structure resembles the configurations found in *Cg-PSII-FCPII* and *PSII-LHCII* from plants. (lines 154-156, revised manuscript).

7. Is FCP-H dimer a homodimer? Please specify.

Author's answers:

*Thank you for raising this issue. It was also an important question puzzling us. Even though some differences were found between loop region residues of FCPII-H1/H2 under the previous poor global and local maps, we assumed that FCPII-H was a homodimer mainly owing to the same protein scaffold and pigment sites (Supplementary Fig. 5i). During this revision, we improved the local map of FCPII-H1/H2 and identified FCPII-H1 as Cm-Lhcf7 and a different FCPII-H2, which was assigned as the homolog of Lhcf7 in *Cyclotella cryptica* (unannotated in the genome). Thus we added a Supplementary Fig. 6 to show the two sequences and their structural differences between FCPII-H1 and FCPII-H2. It can be seen that the counterpart amino acids F89_{FCPII-H1}/T76_{FCPII-H2}, A114_{FCPII-H1}/Y101_{FCPII-H2} and V132_{FCPII-H1}/Y119_{FCPII-H2} of the two FCP subunits are remarkably different. Therefore, we define FCPII-H1/H2 as a heterodimer in the new manuscript and PDB file.*

8. Line 150-153. Please clarify whether *C. meneghiniana* does not contain the PsbG gene, or the PsbG protein is present in but detaches from the supercomplex.

Author's answers:

*Thank you for your question. PsbG is a small, single-transmembrane helix subunit that was newly identified in the structure of the diatom *C. gracilis* PSII-FCPII. However, the sequence of PsbG was not determined, and it was assigned as poly-alanines in the two previously published structures of PSII-FCPII. Therefore, we cannot confirm whether there is a similar PsbG sequence in the *C. meneghiniana* transcriptome. We think it is necessary to determine the PsbG gene sequence in further research to gain more insights regarding this point. At least in the present map, we did not find a *Cg*-PsbG like subunit at the corresponding position (Supplementary Fig. 4a). This may be related to species variations of diatoms in order for the binding of different peripheral FCPs, such as monomer, dimer, trimer or tetramer. To make this point more clearly, we clarified that PsbG is not present in *Cm*-PSII-FCPII supercomplex in the legend of Supplementary Fig. 4a.*

9. Line 174-175. It is evident that the three HPLC curves in Fig. 1f were normalized to the Chl a content. However, PSII-FCPII contains much more Chl a than FCP trimer and dimer due to the large amount of Chl a bound to the core subunits. Therefore, a lower peak of Chl c in PSII-FCPII sample does not necessarily mean that it binds very few Chl c.

Author's answers:

*Thank you for your comments, and we agree with your viewpoint on the amount of Chl c in the PSII-FCPII sample. Usually, Chls c bind to several specific pigment sites of FCP antennae according to the previous structural results. Because of limited local density map and absence of Fx306/307 as a marker as seen in *Pt-Lhcf4* and *Cg-Lhcf1*, we cannot identify any Chl c sites based on the present local density map of FCPII-G/H/I/J/K. We found the typical phytol tail of Chl a in many 403, 405*

sites of the FCPII-G/H1/H2 antennae, indicating that these Chls *c* sites in the Cm-FCP trimer were changed to Chls *a* in FCPII-G/H1/H2. Moreover, the conserved Lhca2 type Cg-FCPII-D and the other two monomeric Cg-FCPII were suggested to contain 1-2 Chl *c* (Pi et al. 2019; Nagao et al. 2022). Thus, we suggest that FCPII-G/H1/H2 bind fewer Chl *c*, and we fit all Chls as Chl *a* in the present Cm-PSII-FCPII structure. Consequently, we modified the sentence in the manuscript, and added the explanation in the Method section: model building and Supplementary Table2:

More Chls *a* and fewer Chls *c* was observed to bind to Cm-FCPII-G/H1/H2 in the present structure, in agreement with the pigment analysis (Fig. 1d, and Supplementary Table 2). However, a higher content of Chl *c* was found in the FCPII dimer and trimer, respectively (Fig. 1d), revealing large differences among the FCP antennae and also suggesting that most Chls *c* of *C. meneghiniana* are associated with the released FCP trimer and dimers. (lines 184-189, revised manuscript).

“It is worth noting that the pigment sites 408 in FCPII-H, 403 in FCPII-G and some sites in FCPII-I/J/K may potentially be assigned as Chls *c*. However, due to the limited resolution of the map, we temporarily assigned them as Chls *a* now” (lines 629-631, revised manuscript).

10. Line 234-235. It is assumed that Fx, Ddx and Dtx are highly similar in their structures. According to Fig. S1f, the resolution of the FCP-H dimer region is around 3.5 angstrom, can the authors distinguish different carotenoids in this region?

Author's answers:

Thank you for your comments. We apologize for not providing a detailed local density map for the peripheral FCPII antennae. We admit that despite a global resolution of PSII-FCPII map at 2.93 Å, the local maps for the peripheral FCPII-H are around 3.5 angstrom. In the revised Supplementary Figure 1, we improved the local resolution of the FCPII-I/J/K region to 3.48 Å, and the FCPII-G/H region to 3.23 Å. The related local maps and models were deposited in the PDB database (codes 8W4O and 8W4P).

As shown in the figure below, the structures of Fx, Ddx, and Dtx are highly similar, but with some differences. Following the better local map of FCPII-G/H, it is possible to identify the longer Fx molecules with an esterified group. However, the difference between Ddx and Dtx is relatively small. Ddx has an additional epoxide group at one head group (highlighted by an ellipse), which results in bigger density in this region. By HPLC analysis, we detected the presence of Ddx and Dtx in this sample. Thus, we examined each carotenoid carefully and found that chain 0 Dtx303 site is suitable for loss of an epoxide group, thus assigned this site to a Dtx molecule. The other Ddx or Dtx sites were all assigned as Ddx owing to uncertainties in the density map.

11. Line 314. C-terminal -> N-terminal

Author's answers:

We apologize for this mistake and have corrected it

12. Line 325. What is “a red Fx304” ?

Author's answers:

Thank you for your comment. "red Fx" refers to specific Fx sites and related polar environments within FCP, giving them red-shifted absorbing properties to harvest solar energy from 500 to 550 nm. We rewrote this sentence and add a reference in the manuscript.

These ligands include a specific “red Fx304” with red-shifted absorption to harvest more green light⁴, which locates horizontally at the stromal surface and enhances green light harvesting of the Cm-FCP05 trimer (Fig. 3b). (lines 313-316, revised manuscript).

13. Line 338-339. According to Fig. 5, Chls 406 and 412 are located at opposite sides of the thylakoid membrane.

Author's answers:

Thank you very much for pointing out this mistake. We corrected “an extra Chl a412” in this sentence to “an extra Chl a415”.

Reviewers' Comments:

Reviewer #1:

Remarks to the Author:

The revised manuscript from Zhao et al, is improved in many ways. The level of image analysis is better, although in my opinion chain w still contains some modeling errors and I also question the choice of focused masks made by the authors, especially in the case of the FCP IJK which includes too much of the PSII signal, this might still limit the resolution of the map around these FCP's. Having said all that, I think the work is now on par with the standard in the field and can be published.

Reviewer #2:

Remarks to the Author:

The authors have addressed my comments adequately, and the manuscript has been greatly improved. I have no further questions.